# A global land aerosol fine-mode fraction dataset (2001–2020) retrieved from MODIS using hybrid physical and deep learning approaches

Xing Yan[1*], Zhou Zang[1], Zhanqing Li[2*], Nana Luo[3], Chen Zuo[1], Yize Jiang[1], Dan Li[1], Yushan Guo[1], Wenji Zhao[4], Wenzhong Shi[5], Maureen Cribb[2]

[1]State Key Laboratory of Remote Sensing Science, College of Global Change and Earth System Science, Beijing Normal University, Beijing, 100875, China
[2]Department of Atmospheric and Oceanic Science and ESSIC, University of Maryland, College Park, MD, 20740, USA
[3]School of Geomatics and Urban Spatial Informatics, Beijing University of Civil Engineering and Architecture, Beijing 102612, China
[4]College of Resource Environment and Tourism, Capital Normal University, Beijing, China
[5]Department of Land Surveying and Geo-Informatics, The Hong Kong Polytechnic University, Hong Kong, China

*Correspondence to*: Xing Yan (yanxing@bnu.edu.cn); Zhanqing Li (zli@atmos.umd.edu)

**Abstract.** The aerosol fine-mode fraction (FMF) is valuable for discriminating natural aerosols from anthropogenic ones. However, most current satellite-based FMF products are highly unreliable over land. Here, we developed a new satellite-based global land daily FMF dataset (Phy-DL FMF) by synergizing the advantages of physical and deep learning methods at a 1° spatial resolution covering the period from 2001 to 2020. The Phy-DL FMF dataset is comparable to Aerosol Robotic Network (AERONET) measurements, based on the analysis of 361,089 data samples from 1170 AERONET sites around the world. Overall, Phy-DL FMF showed a root-mean-square error (RMSE) of 0.136 and correlation coefficient of 0.68, and the proportion of results that fell within the ±20% expected error (EE) envelopes was 79.15%. Moreover, the out-of-site validation from the Surface Radiation Budget (SURFRAD) observations revealed that the RMSE of Phy-DL FMF is 0.144 (72.50% results fell within the ±20% EE). Phy-DL FMF showed superior performance over alternate deep learning or physical approaches (such as the spectral deconvolution algorithm presented in our previous studies), particularly for forests, grasslands, croplands, and urban and barren land types. As a long-term dataset, Phy-DL FMF is able to show an overall significant decreasing trend (at a 95% significance level) over global land areas. Based on the trend analysis of Phy-DL FMF for different countries, the upward trend in the FMFs was particularly strong over India and the western USA. Overall, this study provides a new FMF dataset for global land areas that can help improve our understanding of spatiotemporal fine- and coarse-mode aerosol changes. The datasets can be downloaded from https://doi.org/10.5281/zenodo.5105617 (Yan, 2021).

## 1 Introduction

Evaluating the impact of anthropogenic aerosols on climate change and human health relies on the ability to separate the proportion of anthropogenic aerosols from the total aerosol loading (Anderson et al., 2005; Zheng et al., 2015; Li et al., 2016). Although satellite remote sensing can provide global-scale data on aerosol content that are represented by the aerosol optical
depth (AOD), accurate monitoring of anthropogenic aerosols is still a major challenge. This is because a key parameter called the aerosol fine-mode fraction (FMF), which is used for discriminating anthropogenic aerosols from natural ones (Bellouin et al., 2005), has been regarded as highly unreliable according to satellite-based AOD retrievals, especially over land (Levy et al., 2013; Yan et al., 2017; Liang et al., 2021; Yang et al., 2020; Zang et al., 2021a).

Satellite-based FMF retrievals based on physical methods have been performed previously; currently, five global-scale
FMF products exist (Figure 1) that exhibit different temporal resolutions from 1 to 16 days (Levy et al., 2007; Garay et al., 2020; Chen et al., 2020a). Of these, POLarization and Directionality of the Earth's Reflectances (POLDER) can perform multi-angle and multi-spectral polarized measurements, which provide unique advantages in the retrieval of aerosol FMF (Dubovik et al., 2011; Dubovik et al., 2019). Therefore, in recent years, several POLDER-based FMF retrieval methods have been proposed (Zhang et al., 2016; Zhang et al., 2021), such as the generalized retrieval of aerosols and surface properties (Dubovik
et al., 2014). However, POLDER ended its mission in 2013, whereas the Moderate Resolution Imaging Spectroradiometer (MODIS) has operated for about 20 years and continues to perform well (K. Yan et al., 2021, G. Yan et al., 2021). In addition, Advanced Along Track Scanning Radiometer (AATSR) ended the mission in 2012 (Kolmonen et al., 2016). While VIIRS started the mission in 2012, which could provide less than 10-year global FMF product so far (Sawyer et al., 2020). Currently, only the MODIS Dark Target (DT) method has been used to generate global aerosol FMF products over both land and ocean.
However, the MODIS DT-derived FMF over land is highly unreliable and is not recommended for use even though it has evolved to the Collection 6.1 (C6.1) level (Levy et al., 2013; Chen et al., 2020a). To improve the accuracy of MODIS land-based FMF retrievals, improvements have been made to physical approaches, such as the Look-Up-Table-based Spectral Deconvolution Algorithm (LUT-SDA) (Yan et al., 2017; Yan et al., 2019). Using the LUT-SDA model in previous research, we developed a 10-year global land FMF dataset (Yan et al., 2021b) with moderately improved retrieval accuracy (root-mean-
square error, RMSE = 0.22). No multi-angle and multi-spectral polarized information, Lipponen et al. (2018) noted that MODIS-based FMF retrievals using physical methods still suffer from these major limitations.

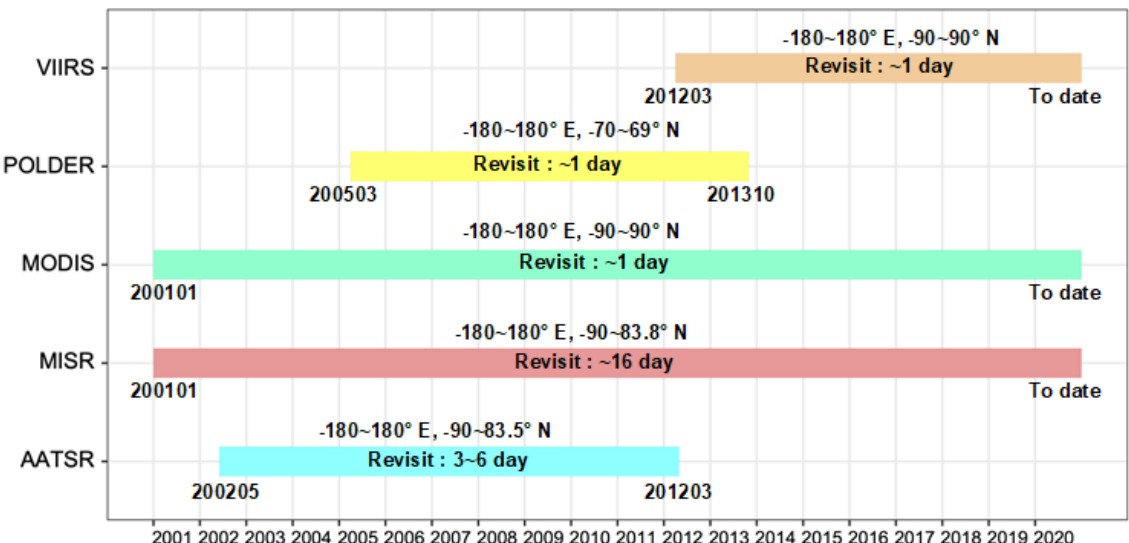

**Figure 1:** Overview of the time periods covered by different satellites that provide global-scale FMF products. Acronyms used in this figure: AATSR: Advanced Along-Track Scanning Radiometer; MISR: Multi-angle Imaging SpectroRadiometer; MODIS: Moderate Resolution Imaging Spectroradiometer; POLDER: POLarization and Directionality of the Earth's Reflectances; VIIRS: Visible Infrared Imaging Radiometer Suite.

In recent years, deep learning approaches have been applied to satellite satellite-based atmospheric research (Zang et al., 2021b; Yan et al., 2020a; Yuan et al., 2020; Shen et al., 2018; Ong et al., 2016), including FMF retrieval (X. Chen et al., 2020). Compared with classical machine learning methods, deep learning is more capable of approximating nonlinear relationships (Yan et al., 2021c). For example, X. Chen et al. (2020) used a convolutional neural network (CNN) to develop a deep learning model for MODIS FMF retrievals called the Neural Network based AEROsol retrieval (NNAero) method. The NNAero-derived FMF is a significant improvement over the MODIS DT-derived FMF, with the RMSE decreasing from 0.34 (DT) to 0.1567 (NNAero). However, this method has only been applied and validated over northern and Eastern China, and not globally. As an important limitation, Zhang et al. (2016) noted that satellite-measured multi-spectral reflectance of ground-based data alone was not sufficient to retrieve FMF with high accuracy. O'Neill et al. (2008) showed that when the temperature is low, the error of the fine-mode AOD calculated by the physical method, i.e., the Spectral Deconvolution Algorithm (SDA), is clearly large (SDA technical memo, O'Neill et al., 2008). Although this issue has long been known, the relationship between meteorological factors and FMF is complex, difficult to describe by equations in the SDA. Benefiting from its powerful ability to describe nonlinear relationships, using a deep-learning model may overcome the deficiencies of the SDA in calculating FMF.

To address the above issues, we synergize the advantages of the physical method and deep learning to retrieve aerosol FMF over land on a global scale using MODIS data. We tested and validated this hybrid model using two decades of data (2001–2020) and produced a new long-term FMF dataset called Phy-DL FMF (physical-deep learning FMF). Contrary to

previous studies, the proposed hybrid model considers both physical characteristics and nonlinear relationships to constrain the FMF calculation. This long-term dataset shows good promise for shedding light on the impacts of human activities on atmospheric aerosols, providing a foundation for understanding the variations in fine mode aerosols on a global scale.

## 2 Materials and methods

### 2.1 MODIS data

The MODIS sensor onboard Terra has provided long-term observations on a global scale every day since February 2000 (Levy et al., 2010), available at the Atmosphere Archive & Distribution System Distributed Active Archive Center. In this study, MODIS C6.1 L1B MOD02SSH data (i.e., top-of-the-atmosphere (TOA) reflectances from Band 1 to Band 7), MODIS C6.1 L3 MOD09CMG data (surface reflectances from Band 1 to Band 7), and MODIS C6.1 L3 MOD08 daily data were 95    obtained from 2001 to 2020 for retrieving FMF. Table S1 summarizes details about the MODIS data used in this study.

### 2.2 AERONET data

The AERONET is a worldwide, sun–sky photometer network providing ground-level aerosol properties, recently updated to Version 3 (Holben et al., 1998). To retrieve FMF from AERONET solar extinction data, O'neill et al. (2001a, 2001b, 2003) developed the SDA method. The FMFs based on this inversion method (i.e., SDA FMF) have been included in the standard 100    AERONET data offering, with an estimated uncertainty of 0.1 (O'neill et al., 2001b; O'neill et al., 2003). Since there is not enough Level 2.0 data for use as training data for modelling purposes, here, we used the Level 1.5 SDA FMF dataset generated from data from 1170 global AERONET sites covering the period of 2001 to 2020 as the ground truth for further modelling and validation (Figure S1a). These AERONET sites are spread around the world, enabling the construction of a universal model and allowing for a more thorough validation of the new FMF product.

### 2.3 Meteorological data

Previous studies have reported that meteorological variables are significantly correlated to fine-mode and coarse-mode aerosols. Tai et al. (2010), Liang et al. (2016), and Shen et al. (2018) all revealed that meteorological variables like temperature, relative humidity (RH), and wind speed explain much of the variations in $PM_{2.5}$ concentrations ($> 50\%$). Xiang et al. (2019) and Gui et al. (2019) found a negative association between Planetary Boundary Layer Height (PBLH) and $PM_{2.5}$, and Kang et 110    al. (2014) found that fine-mode aerosols and air pressure were correlated significantly. In this study, to investigate the correlation between meteorological variables and the FMF, we implemented the Generalized Additive Model (GAM). Figure S2 reveals that the meteorological variables considered in this study, i.e., PBLH, temperature, surface pressure, RH, and wind speed, all had significant non-linear relationships with the FMF (at the 99% significance level). Both PBLH and surface pressure had similar influences on the FMF, i.e., a positive (negative) response when PBLH and surface pressure values were

low (high). This is because high PBLH and surface pressure values can increase the diffusion of fine particles, decreasing the magnitude of the FMF (Tai et al., 2010). Meanwhile, the negative response of the FMF to wind speed also reflects the influence of fine-particle diffusion, as well as the contribution of dust particles strengthened by wind speed (Luo et al., 2016). Increasing temperatures corresponded to decreasing FMFs, partly due to unfavorable diffusion conditions (Tai et al., 2010). On the other hand, more fine particles are released by heating during colder seasons than during warmer seasons (Ramachandran, 2007). RH had a strong positive influence on $PM_{2.5}$ concentrations when RH was between 25% and 75%. This reflects the secondary particle formation boosted by the increasing RH that contributed to the fine particles (Tai et al., 2010). Therefore, in this study, we used surface temperature, air pressure, PBLH, RH, and wind speed as inputs to the deep-learning model.

Due to the impact of meteorological factors on FMF, five meteorological variables (i.e., 2-m air temperature, planetary boundary layer height, surface pressure, 10-m U/V wind components, and 2-m dew point temperature) were obtained from the fifth-generation product produced by the European Centre for Medium Range Weather Forecasts (ERA5), with hourly data available since 1950 and at a 0.25 °spatial resolution (Figure S1b-f). The RH was then calculated by 2-m dew point temperature and air temperature (Tetens, 1930). Given the overpass time and spatial resolution of MODIS data, only monitoring-time meteorological data collected from 10:00 to 11:00 local time were used and resampled to 1 °×1 °to obtain daily averages.

**2.4 Combining physical and deep learning models (Phy-DL) for retrieving FMF**

In this study, we used a "concatenation" mode to combine a physical model and a deep-learning model, i.e., the outputs of the physical model were used as the inputs for the deep-learning model (Figure 3). The physical model used is the LUT-SDA (Yan et al., 2017). The LUT-SDA is designed for satellite FMF retrievals when only AODs at two wavelengths are available (such as DT AOD products). As shown in Eq. (1) of the SDA (O'Neill et al., 2001a), a minimum of AODs at three wavelengths are needed to first obtain the AE derivative ($\alpha'$). The AE of the fine-mode AOD ($\alpha_f$) and the FMF can then be calculated.

$$\begin{cases} \alpha_f = \dfrac{1}{2(1-a)} \ \{(\alpha-\alpha_c - \dfrac{\alpha'-\alpha_c'}{\alpha-\alpha_c} + b^*) + [(\alpha-\alpha_c - \dfrac{\alpha'-\alpha_c'}{\alpha-\alpha_c} + b^*)^2 + 4c^*(1-a)]^{1/2}\} + \alpha_c \\ \\ \text{FMF} = \dfrac{\alpha-\alpha_c}{\alpha_f-\alpha_c} \end{cases} \quad (1)$$

where a, b*, c*, $\alpha_c'$, and $\alpha_c$ are fixed parameters described in section 1 of the Supplementary Information document, based on O'Neill (2010). Since AODs at two wavelengths are not sufficient to calculate $\alpha'$, for the global physically based FMF retrieval, we first divide the whole world into nine regions [as done by Sayer et al. (2014)] and use historical AERONET observed data to determine $\alpha'$ value ranges in these regions. The $\alpha'$ range of values is based on the first and third quartiles of AERONET measurements in different seasons. For example, in Southeast Asia, $\alpha'$ ranges from 0.12 to 0.60 in spring (Yan et al., 2021). In these nine regions, a set of hypothetical values for $\alpha'$ [as determined by Yan et al. (2021)], $\alpha_f$, and AE ($\alpha$) are imported into the SDA [Eq. (1)] to build the relationship with FMF (Figure 2).

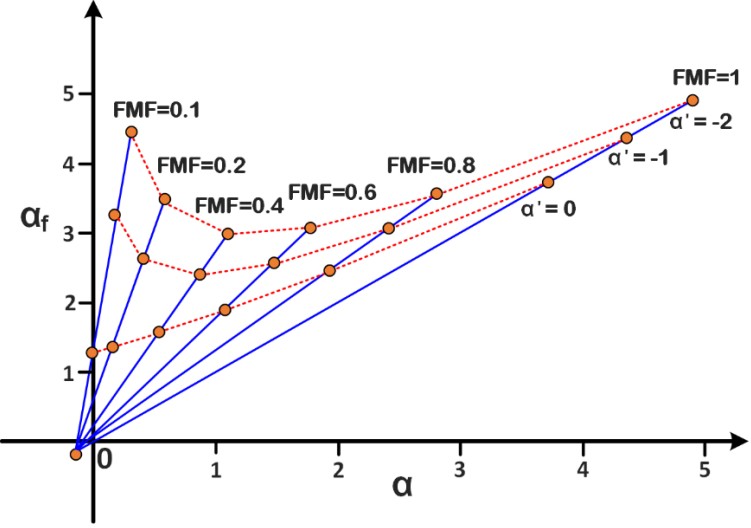

**Figure 2:** Visual representation of SDA-based FMF retrieval LUT.

Different LUTs based on the SDA for these regions are thus created. Based on the constructed LUT, initial results are obtained using a cost function:

$$(FMF^1, \alpha'^1, \alpha_f^{\;1}) = \min[(LUT - SDA_{AE} - MODIS_{AE})]^2 \qquad (2)$$

where $FMF^1$, $\alpha'^1$, and $\alpha_f^{\;1}$ are uncorrected initial results of FMF, $\alpha'$, and $\alpha_f$ by the LUT-SDA, $LUT - SDA_{AE}$ is the $\alpha$ in the LUT, and $MODIS_{AE}$ is the MODIS MOD08 DT-based AE. After performing the $\alpha'$ bias error correction [described in Supplementary Information, Section 2, O'Neill et al. (2003)] and the mean of extreme (MOE) modification [described in Supplementary Information, Section 3, O'Neill et al. (2008)], the final FMF output is:

$$FMF_{output} = \frac{\alpha - \alpha_c}{\alpha_{fcorrected}^1 - \alpha_c} \qquad (3)$$

The deep learning model used in this study is called EntityDenseNet (Yan et al., 2020). The EntityDenseNet incorporates the Entity Embeddings method (Guo and Berkhahn, 2016) that can directly process spatial or time-based features, such as location, season, and month. It includes one input layer, two hidden layers, and one output layer. Each hidden layer has one fully connected layer, one rectified linear unit (ReLU) layer, one batch normalization (BN) layer, and one dropout layer. The feed-forward operation of each hidden layer can be written as

$$a^{n+1} = BN\{f[W^{n+1} D(a^n) + b^{n+1}]\} \qquad (2)$$

where $n$ is the layer number, $a^n$ is the output vector from layer $n$, D() is the dropout layer for the thinning vector $a^n$, $W^{n+1}$ and $b^{n+1}$ are weights and biases, respectively, at layer $n+1$, $f[]$ is the ReLU activation function, and BN is the batch normalization function.

In this study, we combine Phy-based FMF into EntityDenseNet along with satellite measurements and meteorological data to reduce FMF retrieval biases (Figure S3). As shown by Yan et al. (2021b), the global land Phy-based FMF is still not reliable enough and there is a room to improve it. Due to its unknown and known error sources (e.g., MODIS-derived AE) and nonlinearity in the data itself, a linear model may not be able to correct these errors. In addition, current physical retrieval methods do not use all the information provided by satellite observations pertaining aerosol size information retrievals (Zang

et al., 2021). Lipponen et al. (2018) showed that applying a machine learning model to satellite TOA reflectance and geometry data can significantly improve the retrieval accuracy of aerosol size . Other studies have also suggested that surface reflectance and meteorological factors can also impact the FMF retrieval accuracy (Yan et al., 2021a; X. Chen et al., 2020). Thus, besides Phy-based FMF, we input MODIS TOA reflectance data, geometry data, surface reflectance, and meteorological data into EntityDenseNet for the final Phy-DL FMF calculation (Table S1). In the deep learning model training process, 70%, 20%, and

10% of all input data are randomly separated into groups of data for training, validation, and testing, respectively. The validation data are used for the hyperparameter optimization (node numbers and dropout rate in each hidden layer) of the deep learning model. The testing data are used to evaluate the performance of the trained deep learning model. When the trained model is finally optimized by the validation and testing data, we apply this trained deep learning model to reconstruct global land FMF for the period of 2001 to 2020.

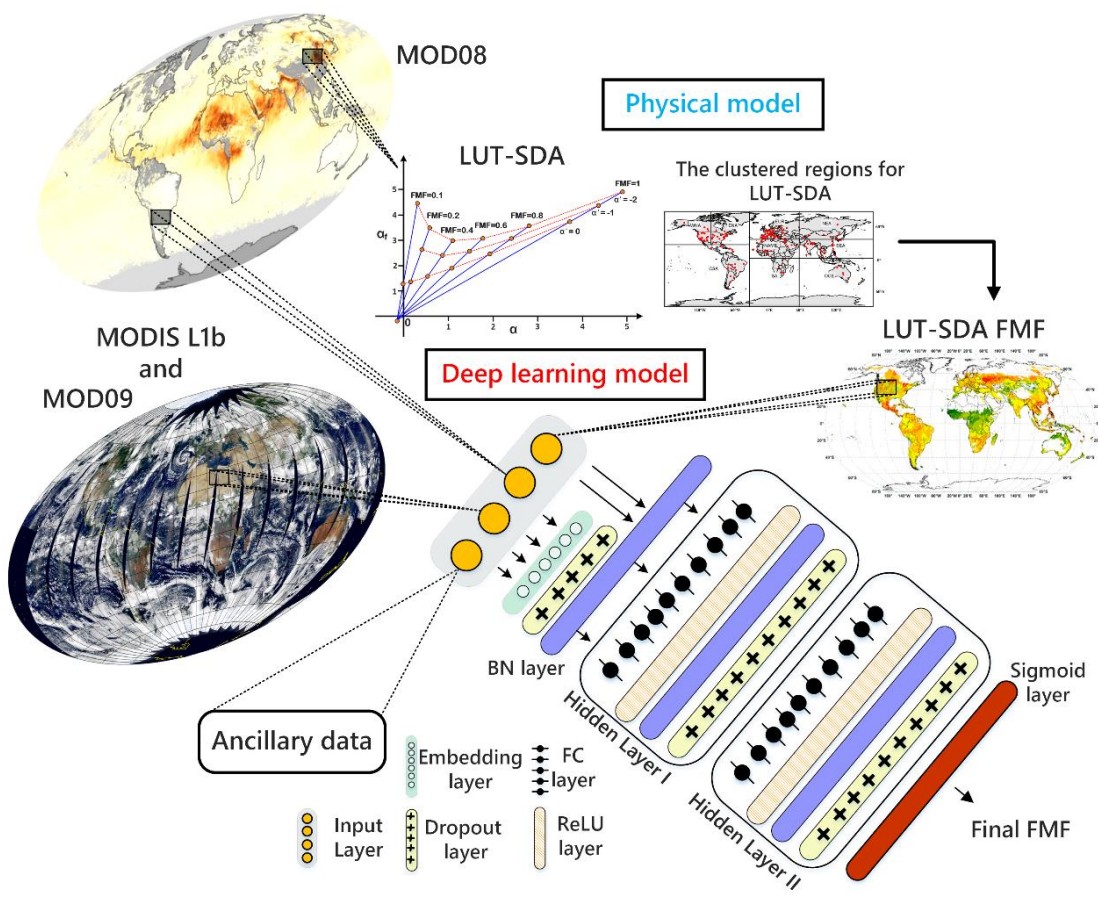

**Figure 3:** Technical flowchart for the production of the global land FMF product.

## 2.5 In-situ observations for independent validation

National Oceanic and Atmospheric Administration's Surface Radiation Budget (SURFRAD) network provides long-term,
multi-band AOD observations at a temporal resolution of three minutes (Augustine et al., 2000). Multifilter Rotating
Shadowband Radiometer (MFRSR) provides spectral solar measurements at SURFRAD sites with approximately 10 nm wide
and the peak nominally at 415, 500, 614, 670, 870, and 940 nm (Harrison et al., 1994). In this study, we selected four
SURFRAD sites (Table S2) which are distant to AERONET sites and applied SDA method to calculate the FMF (SURFRAD
FMF) for validation purpose. Because SURFRAD FMF was not included in modeling training, this data is suitable as the
independent validation for FMF products.

## 2.6 Other global FMF products for comparison

Phy-DL-derived FMFs were compared with the following FMF products from three other satellite missions (Table S3):

a.  POLDER/GRASP FMF:

Launched in December 2004, POLDER-3 onboard the Polarization and Anisotropy of Reflectances for Atmospheric Sciences coupled with Observations from a Lidar satellite was operational from March 2005 to October 2013, making multi-angular polarization measurements. By capitalizing on the small and fairly neutral polarized reflectances (Deuze et al., 2001), POLDER/GRASP is able to provide the fine-mode AOD (fAOD, radius < 0.35 μm) in two categories: "high-precision" and "models". Because "high-precision" fAODs perform better than "models" fAODs (Wei et al., 2020), we used monthly "high-precision" POLDER/GRASP fAODs and AODs (both at 490 nm) at a spatial resolution of 1 °for calculating FMF (at 490 nm) (FMF=fAOD/AOD).

b.  Multi-angle Imaging SpectroRadiometer (MISR) FMF:

The MISR instrument onboard the National Aeronautics and Space Administration Earth Observing System Terra satellite has been continuously working since 2000 (Diner et al., 1998; Kahn and Gaitley, 2015). MISR has nine push-broom cameras with different nominal viewing angles, allowing it to distinguish aerosol type, including aerosol size (Garay et al., 2020). The MISR algorithm retrieves small-mode AODs (at 550 nm) due to aerosol particles with radii less than 0.35 μm at a spatial resolution of 0.5 °. We used it to calculated FMF as the ratio of fAOD and AOD for further comparison in this study.

c.  MODIS FMF:

The latest C6.1 MODIS aerosol product (Levy et al., 2013) no longer includes global-scale FMF, so we used FMF at 550 nm from the previous collection (C5) for comparison purposes (Levy et al., 2007). Although this MODIS FMF product is not reliable over land (Levy et al., 2010), it was used in numerous previous studies (Ramachandran, 2007; Vinoj et al., 2014) including for $PM_{2.5}$ estimations (Li et al., 2016; Zhang and Li, 2015).

## 3. Results

### 3.1 Phy-DL FMF validation

Figure 4 shows the validation of the Phy-DL FMF against AERONET FMF. By matching 20 years of estimated Phy-DL FMF with AERONET FMF (number of match-ups, N = 361,089), we first evaluated the overall performance of Phy-DL FMF (Figure 4a). The correlation coefficient (R) was 0.68, and the root-mean-square error (RMSE) was 0.136. Approximately 90% and 79% of retrievals fell within the expected error (EE) envelopes of ±40% and ±20%, respectively (these envelopes have been adopted from X. Chen et al. (2020)). Figure 4b shows the biases of the Phy-DL FMF (estimated FMF minus AERONET FMF) as a function of the AERONET FMF. The Phy-DL FMF slightly underestimated the FMF, with a negative median bias in each FMF bin. As each FMF increased, a higher percentage of retrievals fell within the ±20% EE envelope, ranging from 42.85% (when FMF < 0.3) to 91.17% (when FMF > 0.8). This indicates that the Phy-DL FMF retrieval performed better when the fine-mode aerosols dominated. Figure 4c shows the validations of Phy-DL FMF over different AERONET sites around the world. Most sites in the eastern USA and Europe have over 70% of Phy-DL FMF falling within the EE envelope of ±20%. Over 90% of Phy-DL FMF fell within the ±20% EE envelope at some sites in the Amazon, southern Africa, and southeast

Asia. However, at coastal AERONET sites in the Caribbean and Mediterranean regions, Australia, and South America, less than 60% of Phy-DL FMF fell within the ±20% EE envelope. A similar result was found for some AERONET sites near deserts in southern South America, Central Asia, Northwest China, and Central Australia.

To further investigate the bias in Phy-DL FMF, Figure S4a shows that more than 75% of the sites located on barren land have low percentages of Phy-DL FMF (< 60%) falling within the EE envelope of ±20%. About 4% of the sites have high percentages of Phy-DL FMF (> 90%) falling within the ±20% EE envelope. This suggests that the accuracy of Phy-DL FMF over barren land is much lower than over other land types. AODs over the bright surface used for the Phy-DL FMF retrieval were significantly overestimated, with the worst performance compared to other vegetated land-cover types (Levy et al., 2010; Petrenko and Ichoku, 2013). This suggests that the performance of the Phy-DL FMF algorithm is poor when applied to regions with barren land. Figure S4b shows the bias of the Phy-DL FMF and the percentage of retrievals falling within the EE envelope of ±20% as a function of Normalized Difference Vegetation Index (NDVI). As NDVI increased from < 0.1 to > 0.8, the percentage of FMF retrievals falling within the ±20% EE envelope also rose from < 70% to > 85%, and the range of bias decreased significantly. The core of the SDA method relies on AE as input (Yan et al., 2017). The AE from the MODIS DT aerosol product is still highly uncertain. The low accuracy of AE can significantly influence the performance of the Phy-DL FMF algorithm. As shown in Figure S5, AEs from the MODIS MOD08 product used as input to the Phy-DL FMF algorithm performed the worst over barren land, with the highest RMSE (> 1) and the lowest percentage of retrievals falling within the EE envelope of ±0.45 (< 45%). This would result in a lower performance of the Phy-DL FMF algorithm when applied to regions with barren land.

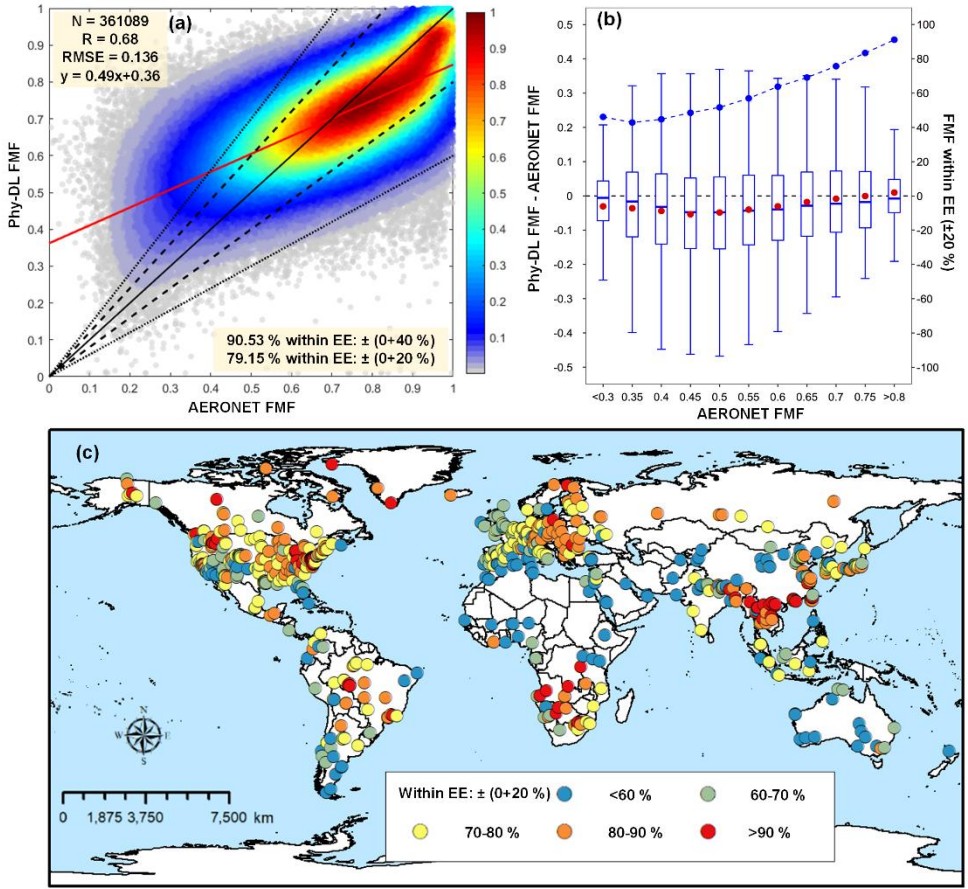

**Figure 4:** (a) Phy-DL FMF at 500 nm as a function of AERONET FMF. The black and red solid lines are the 1:1 line and the best-fit line
obtained from linear regression, respectively. The black dashed and dotted lines represent the expected error (EE) envelopes of ±20% and
±40%, respectively. (b) Box plots of the FMF bias (estimated FMF minus AERONET FMF) as a function of AERONET FMF. The black
horizontal dashed line indicates the zero bias. The red dot in each box represents the mean value of the FMF bias. The upper, middle, and
lower horizontal lines in each box show the 75th, median, and 25th percentiles, respectively. The blue dots connected by the dashed curve
are percentages of FMF retrievals falling within the EE envelope of ±20%. (c) Global distribution of percentages of Phy-DL FMFs falling
within the EE envelope of ±20% at the AERONET sites.

Four sites from the SURFRAD network were selected for the independent validation of the Phy-DL FMF algorithm. As
shown in Figure 5a, the four sites (black triangles) are located across the US, covering different land types from forested land
to barren land. Figure 5b shows how SURFRAD and Phy-DL FMF compare. The R was 0.51, and the RMSE was 0.144,
somewhat different than AERONET validation results (i.e., R=0.68 and RMSE=0.136). Furthermore, the Phy-DL FMF
performance was validated at each SURFRAD site. Figure 5c shows the bias of Phy-DL FMF (Phy-DL FMF minus SURFRAD
FMF), percentage of retrievals falling within the ±20% EE envelope, and RMSEs at each site. In general, most of the sites

have a mean bias and an RMSE lower than 0.1 and 0.15, respectively, with over 70% of the retrievals falling within the ±20% EE envelope. The out-of-site validation reveals that the Phy-DL FMF algorithm is reliable even in regions without AERONET sites for model training.

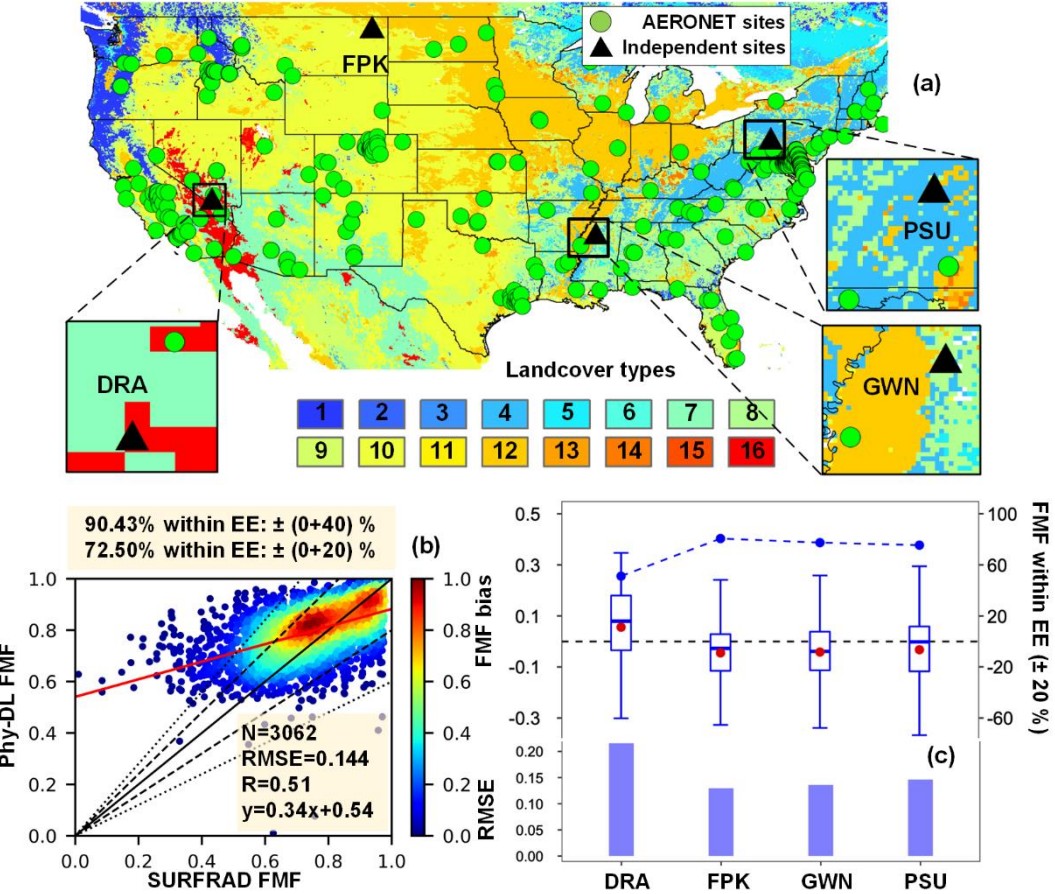

**Figure 5:** (a) The locations of AERONET sites (green points) and four independent SURFRAD sites (black triangles) for the independent validation of the Phy-DL FMF algorithm. The base map shows the land types from MODIS MCD12C1 data (the International Geosphere-Biosphere Programme scheme, Table S4). (b) Phy-DL FMF at 500 nm as a function of SURFRAD FMF. The black and red solid lines are the 1:1 line and the best-fit line obtained from linear regression, respectively. The black dashed and dotted lines represent the expected error (EE) envelopes of ±20% and ±40%, respectively. (c) Boxplots of bias (Phy-DL FMF minus SURFRAD FMF), percentage of FMF estimates falling within the EE envelope of ±20% (dash-dotted lines), and RMSEs at the four independent SURFRAD sites. The upper, middle, and lower lines in each box present the 75th, median, and 25th percentiles, respectively. The red point in each box represents the mean value of the FMF bias.

Figure S6 shows the frequencies of three FMF levels (low: FMF < 0.5, medium: 0.5 < FMF < 0.8, high: FMF > 0.8, Supplementary section S4) based on Phy-DL and AERONET FMF data from 2001 to 2020. Over 60% of AERONET-derived

FMFs were low over Central Asia, Central Australia, and the sub-Sahara; the AOD of these locations was dominated by coarse-mode aerosols (dust). Phy-DL-estimated FMFs were also low over Central Asia and the sub-Sahara, but slightly underestimated over Central Australia (frequency < 40%). Over 90% of both AERONET and Phy-DL FMFs were at the medium level in South America, Western Africa, Australia, Western Asia, and the Western USA. Approximately 45–55% of Phy-DL and AERONET FMFs were at the medium level in the Eastern USA, Europe, and Central Africa. Over 60% of Phy-DL and AERONET FMFs were at a high level in Northern India, Southeast Asia, and Southeast China.

## 3.2 Global distribution of FMF over land and trends from 2001 to 2020

Figure 6a shows the global distribution of mean Phy-DL FMF over land from 2001 to 2020. A high proportion of fine-mode aerosols with FMF greater than 0.77 can be seen in populated regions, including Southern China, Southeast Asia, Eastern Europe, and the Eastern USA. Low FMF values (< 0.55) were observed in Australia, Northwest China, Central Asia, the Saharan region, southern South America, and the Southeastern USA, where coarse-mode aerosols from large deserts dominate. Figure 6b shows the spatial distributions of the Phy-DL and AERONET FMFs linear trends from 2001 to 2020. In general, both datasets show decreasing trends [i.e., < $-3 \times 10^{-3}$ year (yr)$^{-1}$] in Northeast China, Central Asia, Europe, the Saharan region, Southern Africa, South America, Mexico, and the eastern USA. In contrast, Southeast Asia, India, Central Australia, Central Africa, and the western USA show significant increasing trends of over $3 \times 10^{-3}$ yr$^{-1}$. The increasing FMF trend over central Australia is sporadic and could be related to an increase in fire activity (Andela et al., 2017). In South America and Africa, the long-term decrease in burning during the past two decades (Andela et al., 2017; Deeter et al., 2018) has contributed to a significant decrease in FMF. However, the reduced biomass burning in Central Africa is also partially offset by the dramatic growth in anthropogenic emissions (Zheng et al., 2019), leading to a slight increasing trend in FMF ($3 \times 10^{-3}$ to $7 \times 10^{-3}$ yr$^{-1}$). The decreasing FMF trends in Europe and the eastern USA are driven by reduced anthropogenic emissions from transportation sources (Crippa et al., 2016; Jiang et al., 2018). The decreasing FMF trend in Northeast China is likely more associated with a decrease in industrial and residential emissions due to the implementation of clean air policies (Van Der Werf et al., 2017; Yang et al., 2018; Zheng et al., 2019). In the western USA, the dramatically increasing FMF trend is likely partly attributed to the increase in smoke from wildfires (Parks and Abatzoglou, 2020; O'dell et al., 2019; Zhang et al., 2020). In India, the significant increase in FMF likely reflects an increase in vehicular anthropogenic emissions and crop residue burning (Jethva et al., 2019; Manoj et al., 2019). Figure 6c shows the time series of the global monthly mean Phy-DL and AERONET FMFs from 2001 to 2020. Both time series show similar annual cycles and decreasing trends (i.e., negative slopes). However, only the Phy-DL-estimated FMF decreasing trend was significant ($-1.9 \times 10^{-3}$ yr$^{-1}$ at 95% significance level). This is because the Phy-DL dataset has greater spatial coverage than that of the point-scale AERONET dataset.

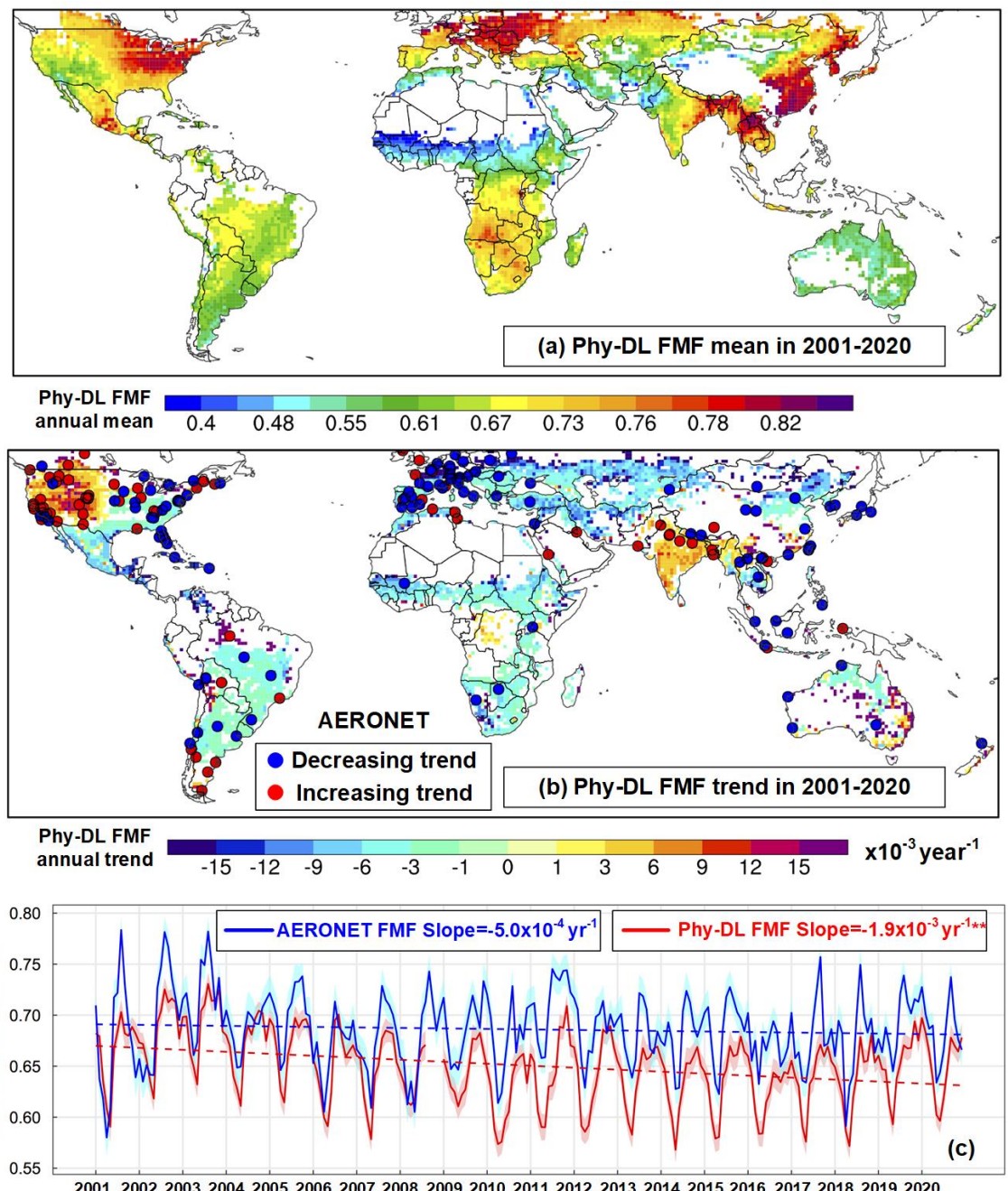

**Figure 6:** (a) Global distribution of Phy-DL FMF mean values over the 2001–2020 period. Only those pixels with over 120 retrievals year‾

¹ (yr⁻¹) were considered. (b) Global distribution of Phy-DL FMF linear trends from 2001 to 2020. Only those pixels with trends at the 95% significance level were considered. The red and blue dots represent AERONET stations with increasing and decreasing linear trends, respectively, at the 95% significance level. (c) Global monthly mean Phy-DL FMF (red line) and AERONET FMF (blue line). The shaded

areas around each line represent the monthly mean FMF value $\pm 0.1 \times$ the monthly standard deviation. The double-asterisks "**" indicate that the linear trend was at the 95% significance level.

Figure 7 shows the global distributions of seasonal Phy-DL-estimated FMFs from 2001 to 2020. In Central Africa, spring had the lowest FMF, especially in northern Central Africa, due to the transportation of dust (Huebert et al., 2003). Meanwhile, FMFs in summer and autumn were higher than those during winter. This is partly attributed to the high temperature and humidity conditions conducive to the formation of fine-mode aerosols (Tan et al., 2015).

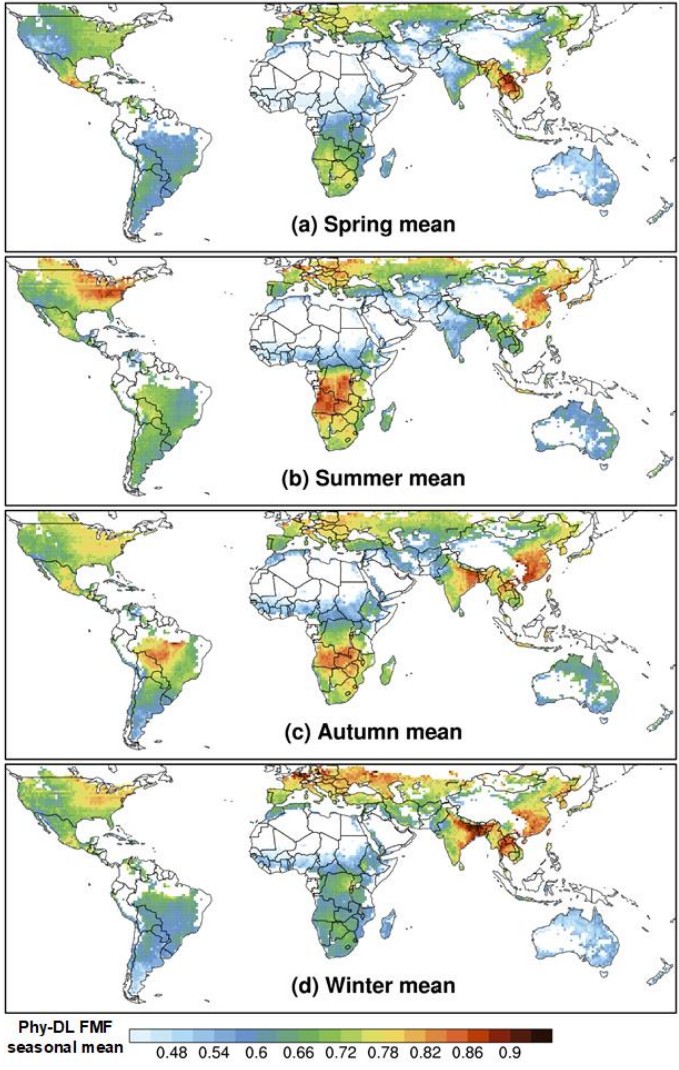

**Figure 7:** Phy-DL-estimated FMF seasonal mean values from 2001 to 2020. The seasons are defined as spring (March, April, and May), summer (June, July, and August), autumn (September, October, and November), and winter (December, January, and February) for both the Northern Hemisphere and Southern Hemisphere. Only those pixels with 120 retrievals yr$^{-1}$ were considered when calculating the mean values.

In India, FMFs were noticeably higher in autumn and winter, especially in Northern India (i.e., the Indo-Gangetic Plain), where the FMF was greater than 0.87. During spring and summer, FMFs were usually less than 0.63 over India. Mhawish et al. (2021) also reported the same seasonal pattern. This is likely related to spring in India being the pre-monsoon season when dust particles from nearby deserts are frequently transported to the country (Gautam et al., 2009). During that season, the dominant coarse-mode aerosols decrease from west to east over the Indo-Gangetic Plain (Kalapureddy and Devara, 2008), thereby leading to lower FMF, particularly in the western Indo-Gangetic Plain. In the post-monsoon seasons (autumn and winter), the higher FMF is attributed to the low boundary layer and non-convective atmosphere, which induces haze and stagnant conditions. Frequent biomass burning events also occur during these seasons (Ramachandran, 2007), which contributes to higher FMFs.

In Central Africa and the Amazon, FMFs in summer and autumn were higher than those in spring and winter (seasons here correspond to the seasons in the Northern Hemisphere). This coincides with local biomass burning, which mainly occurs from early summer to the middle of autumn (Generoso et al., 2003; Perez-Ramirez et al., 2017). Accordingly, fine-mode aerosols including black carbon and organic carbon can contribute to higher FMFs in summer and autumn. Although Australia had low FMFs (< 0.6) in all seasons, some sporadic pixels in autumn had FMFs near 0.7, which may also be related to the frequent wildfires in autumn (Shi et al., 2021; Liu et al., 2021).

In the eastern USA, FMFs were the highest in summer; however, in the western USA, FMFs were the highest in winter. Across the entire USA, FMFs were lowest in spring. In the eastern USA, it is thought that accelerated photochemical reactions and stagnant conditions in summer produce the highest amount of ammonium sulfate in all seasons (Tai et al., 2010). Moreover, ammonium nitrate is the main component of fine-mode particles in the western USA whose content peaks in winter (Hand et al., 2012). This explains why FMF maxima occur in different seasons on both sides of the country.

In Eastern China, summer and autumn had higher FMFs (> 0.8) than those in spring and winter (< 0.78). This is probably because warm seasons with relatively high humidity and temperature can enhance the generation of secondary fine particles by gas-to-particle conversions (Tan et al., 2015). In addition, springtime dust transportation in Northeastern China results in increasing coarse dust particles, thereby affecting the FMF (Huebert et al., 2003). In contrast, Southeastern Asia had exceedingly higher FMFs in winter and spring (> 0.86) than those in summer and autumn (< 0.8), owing to the intense biomass burning from January to April (Yin et al., 2019).

### 3.3 Comparison between Phy-DL, DL-based, and Phy-based FMFs

To analyze the differences in FMFs obtained by different methods, FMFs generated by the Phy-DL method, deep learning (DL) method (meaning no Phy-based FMF as input), and Phy-based method (i.e., the LUT-SDA) from 2008 to 2017 were compared using AERONET FMF as the ground truth. Figure 8a shows the three types of FMF estimates that were averaged into 20 bins with AERONET measurements AOD >0.2 based on the method in Levy et al. (2007). Compared with Phy-based FMF, DL-based FMF has better estimation for low FMF (<0.6), showing the overall improvement in R from 0.51

to 0.60. However, there is still significant underestimation for DL-based FMF when AERONET FMF is greater than 0.6. Phy-DL FMF ameliorated the retrievals by reducing both the underestimation for high FMF values and overestimation for low
FMF values, with the highest R (0.81) among three FMFs. The regression equation of Phy-DL FMF also improved tremendously, with smaller intercept and slope closer to 1.

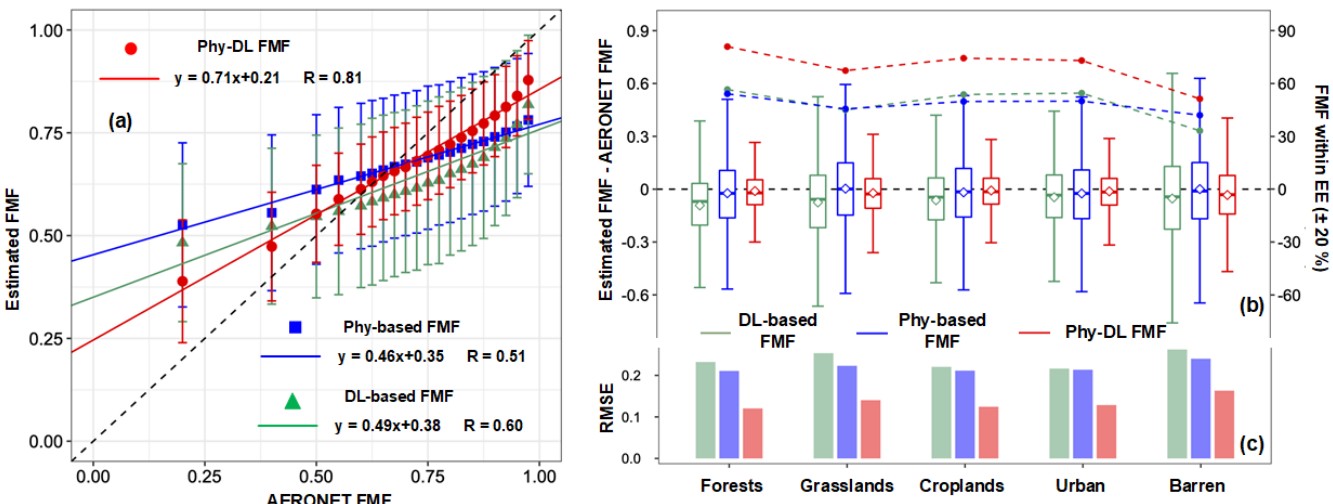

**Figure 8:** Phy-DL (red), Phy-based (blue), and DL-based FMF (green) estimation compared with AERONET FMFs for AOD > 0.2 (at 500
360    nm, using data from 2008 to 2017). (a) The dots and the error bars indicate the means and standard deviations of the FMF estimates in 20 bins of AERONET FMF. The solid blue and red lines are the best-fit lines from linear regression. The black dashed line represents the 1:1 line. Linear regression relations and correlation coefficients (R) are given. (b) Boxplots of bias (estimated FMF minus AERONET FMF) and percentage of FMF estimates falling within the EE envelope of ±20 % (dotted, dashed lines) as a function of land type. The upper, middle, and lower lines in each box presents the 75th, median, and 25th percentiles, respectively. The diamond in each box represents the
mean value of the FMF bias. (c) The RMSE for each land type against that of the AERONET FMF.

Figures 8b and 8c compare the accuracy between Phy-DL, Phy-based, and DL-based FMFs over five land types (forests, grasslands, croplands, urban, and barren). The five land types were selected based on MODIS MCD12C1 data from the International Geosphere-Biosphere Programme scheme. Figure 8b shows that Phy-DL FMF had the lowest bias with mean
values close to 0, smallest range of bias, and highest FMF retrievals (within ±20% EE) over all land types. Although DL-based FMF had a slightly smaller range of bias and higher FMF retrievals (within ±20% EE) than those of Phy-based FMF over forests, croplands, and urban land types, DL-based FMF still had the largest mean bias and showed the worst performance over barren land types. In addition, the DL-based FMF had the highest RMSE among all the FMFs for all land types. Figure 8b shows that Phy-DL, Phy-based, and DL-based FMFs all had the best performance over forests, with RMSE values of 0.120,
0.211, and 0.223, respectively (Figure 8c). Likewise, all performed the worst over barren land, showing a significant negative bias, with less than 50% of the FMF retrievals falling within the ±20% EE envelope. Overall, Phy-DL-estimated FMFs showed

a significant improvement over the Phy-based and DL-based FMFs, especially over forests, croplands, and urban land types, where the RMSEs and biases were noticeably reduced.

For further evaluation, Phy-based, DL-based and Phy-DL FMF were validated against AERONET FMF over AERONET sites to show their spatial performance (Figure S7). DL-based FMF have generally the highest RMSE, with 93.2% sites having RMSE greater than 0.11, compared to 81.0% sites for Phy-based FMF and only 34.8% sites for Phy-DL FMF. Especially, in Australia, India, southern South America, Mediterranean region and North America, DL-based FMF has RMSE dominantly exceeds 0.23 but the RMSE of Phy-based FMF ranges from 0.11~0.23 and Phy-DL FMF is lower than 0.17. Phy-DL FMF performed well in eastern Asia, southern Africa, Europe and eastern US, with RMSE typically lower than 0.11. In contrast, Phy-based FMF in these regions has RMSE greater than 0.11, and DL-based FMF even has large amount of site with RMSE over 0.23. Regarding to R, 69% sites of Phy-DL FMF have R over 0.6, but only 21% sites of Phy-based FMF and 11% sites of DL-based FMF reach R over 0.6. According to Figure S7b, d and f, although DL-based FMF has fewer sites with R less than 0.1 in Europe and North America than Phy-based FMF, there are limited sites for both FMFs in Eastern China, India, southeastern Asia, Saharan region and eastern US having high R (>0.6). However, most of sites for Phy-DL FMF achieves this high R.

Figure 9 compares the annual mean FMF from 2008-2017 based on Phy-based, DL-based and Phy-DL FMF estimations. In general, high FMFs (> 0.7) were well captured by both estimation methods over Eastern China, Southeast Asia, Europe, Southern Africa, the eastern USA, and Mexico. However, compared to DL-based and Phy-DL FMF, Phy-based FMF tends to underestimate the hotspots of FMF, such as Eastern China and central Africa. While in some regions with comparatively low FMF (<0.55), the estimations also show large differences. For example, in northeast Australia and southern South America, Phy-DL and AERONET FMFs agreed well with values less than 0.55, but Phy-based FMFs were clearly overestimated by ~0.1. In addition, in regions dominated by coarse mode aerosol such as Saharan region and central Asia, only Phy-DL FMF captured this low FMF (<0.45), while Phy-based FMF shows overestimation by ~0.1. DL-based FMF also captured the low FMF in central Asia, yet overestimated the FMF in Saharan region. In central Africa, FMF value is relatively high (>0.7) according to AERONET. Phy-DL and DL-based FMF captured this high value yet Phy-based FMF is greatly underestimated, with FMF less than 0.7. In Australia, only Phy-DL FMF well agreed with AERONET FMF of values less than 0.6, while both DL-based and Phy-based FMF show severely overestimations with FMF values reach over 0.65.

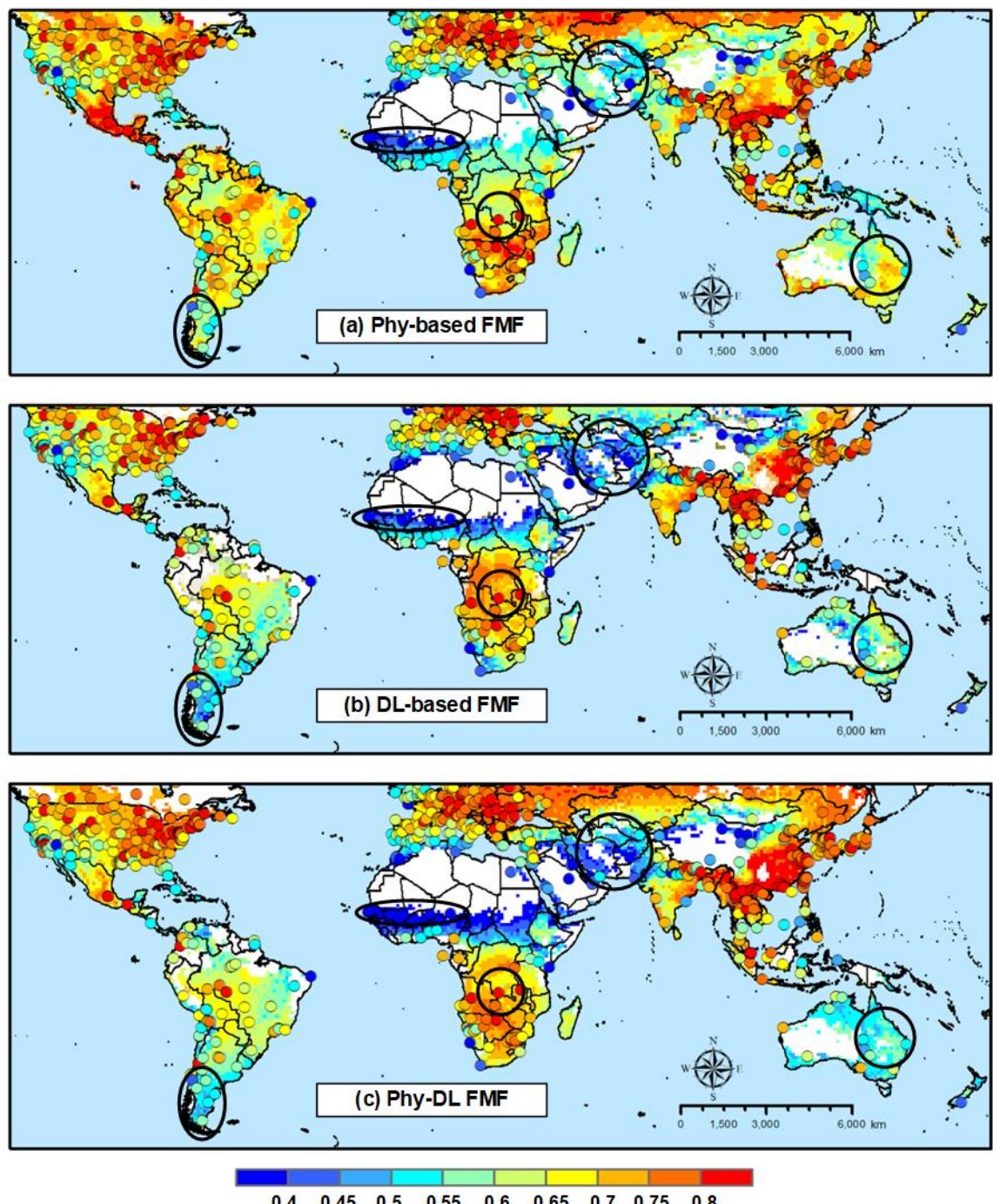

**Figure 9:** Annual mean (a) Phy-DL FMF estimates, (b) Phy-based FMF estimates, and (c) DL-based FMF estimates. The colored dots in (a), (b), and (c) show annual mean AERONET FMF. Areas outlined in black show regions with noticeably large differences in the FMF estimates. Only those pixels with over 120 retrievals yr$^{-1}$ were considered. Data from 2008 to 2017 were averaged.

## 3.4 Comparison with other satellite-based FMF products

Figure 10a–d shows the performance of Phy-DL, POLDER, MISR, and MODIS FMFs against AERONET FMFs. Because these three satellite FMF products cover different time ranges, we only compare retrievals made during the overlapping period from 2008 to 2013 when all products were available. The Phy-DL FMF performed the best, with R and RMSE values of 0.78 and 0.100, respectively. In addition, 96.31% (84.74%) of Phy-DL FMF fell within the EE envelope of ±40% (±20%), an improvement over other FMF products. The next best-performing FMFs were from POLDER and MISR.

POLDER shows R and RMSE values of 0.48 and 0.233, respectively, and 76.05% (46.99%) of the retrievals falling within the EE envelope of ±40% (±20%), while MISR FMF has R and RMSE of 0.42 and 0.204, and 85.01% (45.85%) retrievals between the EE envelop of ±40% (±20%). Both POLDER and MISR FMFs were underestimated compared to AERONET FMF, especially when the AERONET FMF was greater than 0.6. In contrast, MODIS FMF was overestimated compared to AERONET FMF, especially for AERONET FMF greater than 0.6, where MODIS FMF reached values near 1. The overall

performance of MODIS FMF was also the worst, with R and RMSE values of 0.37 and 0.282, respectively, and 68.88% (44.48%) of the retrievals falling within the EE envelope of ±40% (±20%). Figure 10e shows the probability density functions (PDFs) of the FMF biases (estimated FMF minus AERONET FMF). The Phy-DL PDF reveals that most of the biases were close to zero, suggesting the robustness of the Phy-DL method. MISR and POLDER PDFs show underestimations, with most of the biases near -0.2 and -0.1, respectively. The MODIS PDF shows overestimations with biases concentrated near 0.05.

Overall, compared with AERONET FMF, of the four FMF products, the Phy-DL-estimated FMF agreed the best. Figure S8 shows the global distributions of RMSE from validations of Phy-DL, POLDER, MISR, and MODIS FMFs against AERONET FMFs at the AERONET sites. Concerning MISR FMF, 47.9% of the sites had RMSEs higher than 0.23, and 5.3% of the sites had RMSEs lower than 0.11, showing the worst performance. Concerning POLDER FMF, 29.7% of the sites had RMSEs higher than 0.23, mainly in the USA, the Amazon, Southern Africa, Western Europe, and Southeast Asia. MODIS FMF

performed well in Eastern China, India, Europe, and the eastern USA, with 40.0% of the sites having RMSEs lower than 0.11. In comparison, the Phy-DL FMF had RMSEs lower than 0.11 for 65.2% of the sites. In addition, the number of match-ups of Phy-DL-estimated and AERONET FMF was the highest (N = 566), indicating a higher data coverage compared with the other FMF products. In terms of R (Figure S9), at 82.2% of the AERONET sites, R for MISR FMF was less than 0.2 (Figure S9c). At 33.8% of the AERONET sites, mainly in Eastern China, India, and Australia, R for MODIS FMF was greater than 0.5, but

at most sites in the USA and Europe, R was less than 0.2 (Figure S9d). At 39.7% of the AERONET sites, R for POLDER FMF was greater than 0.5 in Europe, the Amazon, and Eastern China, but at most sites in the USA, India, and Australia, R was less than 0.2 (Figure S9b). The R for Phy-DL FMF was greater than 0.5 at 79.0% of the AERONET sites, agreeing better with AERONET FMF than did POLDER and MODIS FMFs in the USA, Africa, Southeast Asia, and Europe (Figure S9a).

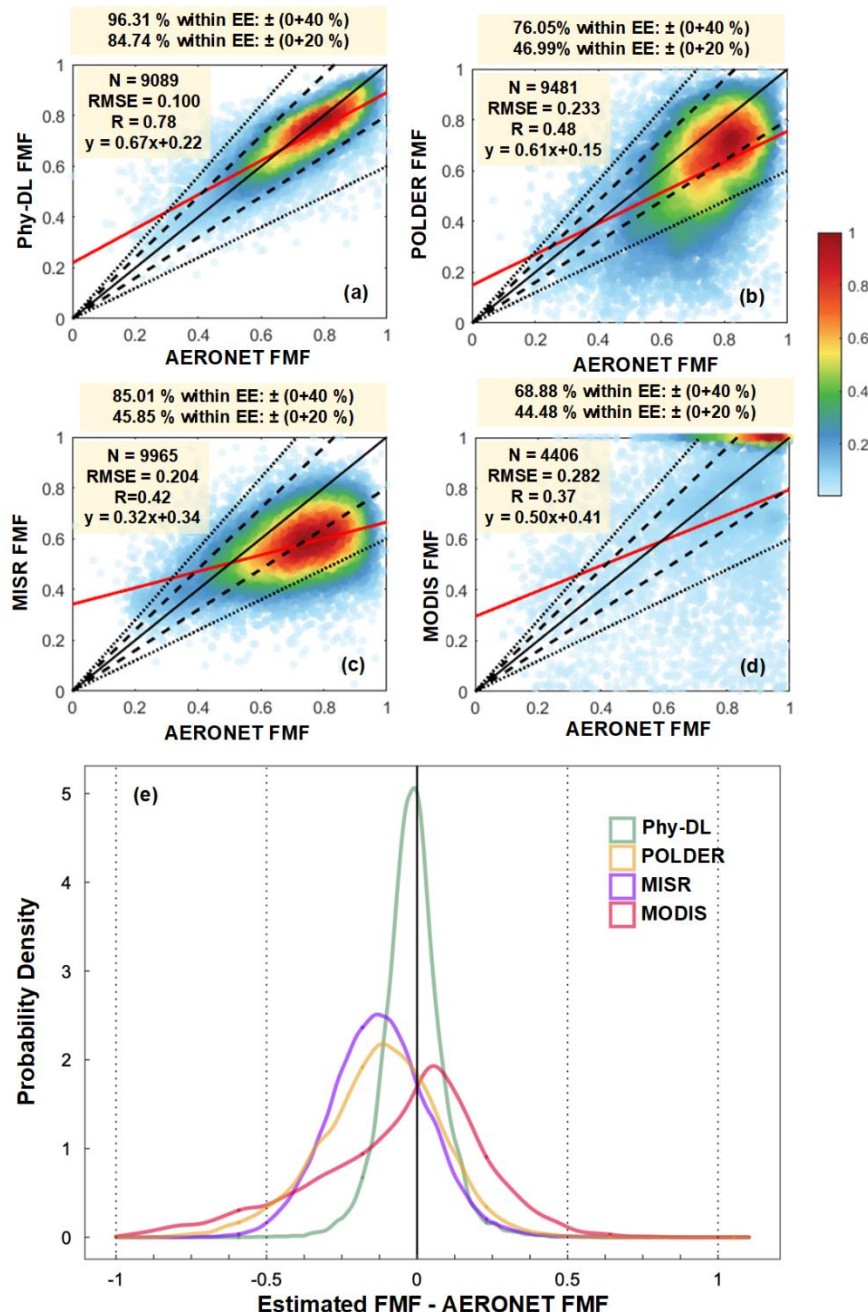

**Figure 10:** Evaluation of (a) Phy-DL (550 nm), (b) POLDER (490 nm), (c) MISR (550 nm), and (d) MODIS FMFs (500 nm) against AERONET FMF (500 nm) from 2008 to 2013. Black and red solid lines are 1:1 reference lines and best-fit lines from linear regression, respectively. Black dashed and dotted lines represent the EE envelopes of ±20% and ±40%, respectively. The number of samples (N), RMSE, correlation coefficient (R), and linear regression relation are given in each panel. (e) Probability density functions of the FMF bias (estimated FMF minus AERONET FMF) for Phy-DL (green), POLDER (orange), MISR (blue) and MODIS (red) FMFs.

The inter-comparison results in Figure S10 shows that when validated by independent FMF observations not used for training in the deep-learning model (SURFRAD FMF), Phy-DL FMF still outperformed the other satellite products, with the highest R (0.51), lowest RMSE (0.143), and the greatest number of retrievals falling within the EE envelopes of ±20% (69.08%) and ±40% (89.05%). PODLER results have an RMSE of 0.232 and R of 0.32, with 76.10% (48.23%) of retrievals falling

within the EE envelope of ±40% (20%). MISR results have an R and RMSE of 0.22 and 0.212, respectively, with 82.61% (45.38%) of retrievals falling within the EE envelope of ±40% (20%). MODIS results were the poorest, with an especially high RMSE (0.465) and low percentages of retrievals falling within the EE envelopes of ±40% (37.23%) and 20% (18.09%). Overall, at the independent SURFRAD sites, Phy-DL FMF is still more accurate and reliable than the other FMF products.

Figure 11 compares the spatial distributions of annual mean MISR-, MODIS-, POLDER-, and Phy-DL-estimated FMFs

from 2008 to 2013. In general, Phy-DL FMF was higher than the satellite-based FMFs over areas of known biomass burning and urban areas, including the eastern USA, the Amazon, Southern Africa, Eastern China, and Australia. Phy-DL and AERONET FMFs in Eastern China reached over 0.7, while POLDER, MISR, and MODIS FMFs were significantly underestimated (~0.6–0.7, ~0.5–0.6, and generally < 0.4, respectively). In the western USA, Phy-DL and AERONET FMFs were higher than 0.6, but MODIS FNFs were < 0.4, and MISR and POLDER FMFs were < 0.6. In Central Africa, POLDER,

Phy-DL, and AERONET FMFs were similar (> 0.7), but MISR FMF ranged from 0.6 to 0.7, and MODIS FMF exceeded 0.8. In Australia and the Amazon, Phy-DL and MISR FMFs agreed well with AERONET FMFs (0.5–0.6 for Australia and ~0.6–0.7 for the Amazon), but POLDER and MODIS FMFs (< 0.4) were significantly underestimated compared with AERONET FMFs. Figure S11 shows the bias, the percentage of FMF retrievals falling within the EE envelope of ±20%, and the RMSEs of MISR, MODIS, POLDER, and Phy-DL FMFs over five land types (forests, grasslands, croplands, urban, and barren), using

data from 2008 to 2013. Over all land types considered and compared with the satellite-based retrievals, Phy-DL FMFs had smallest biases, a higher percentage of FMFs falling within the EE envelope (> 67%), and the lowest RMSE (< 0.127). Both POLDER and MISR FMFs had significant negative biases of -0.2 and -0.1, respectively, over all land types. MODIS FMF had significant positive biases over forests and grasslands and negative biases over croplands, urban areas, and barren areas. Over forests, grasslands, croplands, and urban areas, MODIS FMF had the largest RMSE (> 0.280), and MISR FMF had the lowest

percentage of FMFs falling within the EE envelope (< 40%). Over barren land and of all FMF products, POLDER FMF was the poorest (23.68% of the FMFs falling within the EE envelope, and RMSE = 0.326).

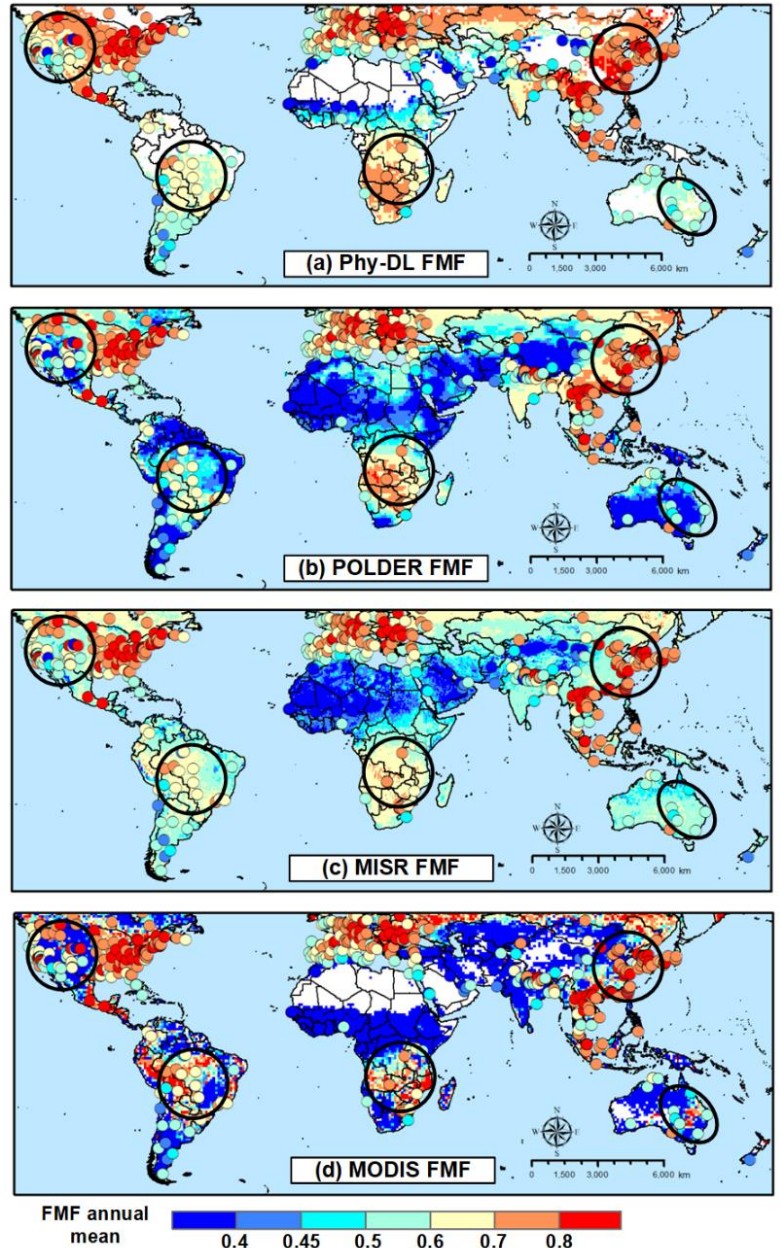

**Figure 11:** Annual mean FMFs based on (a) Phy-DL, (b) POLDER, (c) MISR, and (d) MODIS. The colored dots show annual mean AERONET FMFs. Areas outlined in black circles show regions with noticeably large differences in the FMF estimates. Only those pixels with over 120 retrievals yr$^{-1}$ were considered in the Phy-DL estimation. Data from 2008 to 2013 were averaged.

Next, we conducted a comprehensive comparison of these satellite-based FMF products over Central Africa. Regarding
FMF annual mean values (Figure 12a–d), the POLDER and Phy-DL FMFs agreed the best with AERONET FMFs, which
captured the high values in the middle part of Central Africa (> 0.76) and the low values along the coasts (< 0.7). Although
MISR FMF also captured the low FMFs over coastal regions, FMFs were underestimated in the interior (< 0.7). However,
MODIS FMF was significantly overestimated along the western coast (> 0.85) and underestimated in the southeastern part of
Central Africa (< 0.4). Linear trends were also calculated for all the FMF products (Figure 12e–h). Note that only the linear
trends significant at the 95% level were examined. AERONET showed a significant increasing trend in the northern part of
Central Africa (+0.01 yr$^{-1}$) and a decreasing trend in the southern region (-0.01 yr$^{-1}$). Of all the FMF products, Phy-DL FMF
trends agreed best with AERONET FMF trends. POLDER and MODIS FMF trends were greatly enhanced in the southern
region (+0.05 yr$^{-1}$), while MISR FMF trends did not reflect the AERONET FMF trends well. Overall, Figure 12 illustrates that
in Central Africa, compared with the three satellite-based FMF products, Phy-DL FMF is more accurate and reliable.

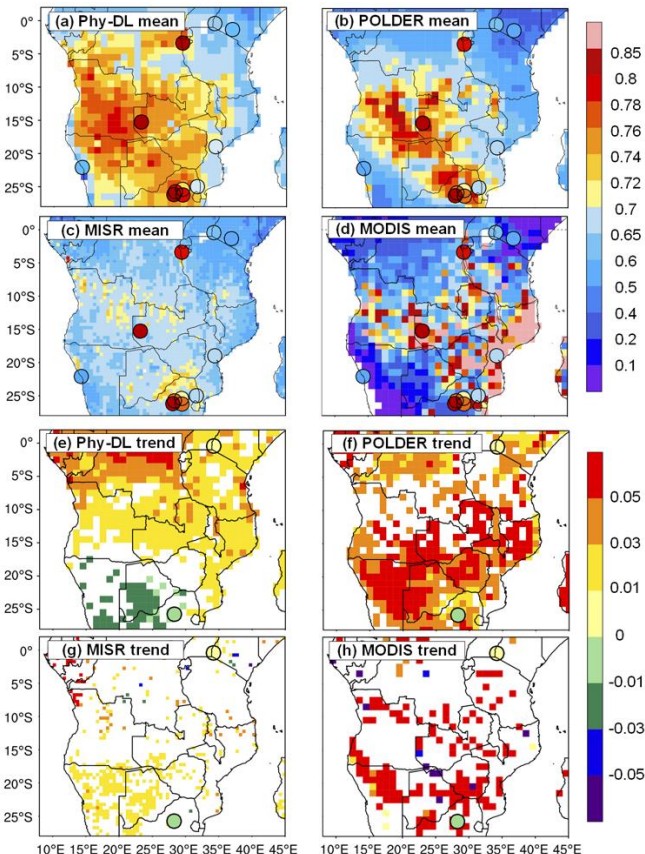


**Figure 12:** (a–d) Spatial distributions of annual mean FMF (averaged from 2008 to 2013) over Central Africa based on Phy-DL, POLDER,
MISR, and MODIS. The colored dots show annual mean AERONET FMF. (e–h) Spatial distributions of the FMF linear trend from 2008 to
2013 over Central Africa based on Phy-DL, POLDER, MISR, and MODIS. The colored dots show linear trends at the AERONET sites.
Only pixels and dots with linear trends at the 95% significance level are shown.

To compare seasonal differences between these methods, Figure 13 compares the seasonal mean Phy-DL-, POLDER-, MISR-, and MODIS- estimated FMFs from 2008 to 2013, and Figure S12 shows their differences (i.e., satellite estimates minus Phy-DL estimates). In all seasons, Phy-DL FMFs were generally higher than MISR FMFs over urban areas and regions where biomass burning was prevalent, such as the USA, Eastern China, and India. During fine-mode-particle-dominated seasons (FMF > 0.8), such as summertime for the eastern USA and wintertime for Eastern China and India, differences between Phy-DL and MISR FMFs reached < -0.18. MODIS FMFs (< 0.2) were much lower than Phy-DL FMFs (> 0.6) in sub-Saharan Africa, India, China, Australia, and the western USA in all four seasons, with differences < -0.5. Conversely, during winters in the Amazon and Central Africa, MODIS FMFs (> 0.74) were slightly higher than Phy-DL FMFs (~0.66); the differences of POLDER FMFs (~0.2) were globally lower than Phy-DL FMFs in all four seasons. In the eastern USA during autumn and winter, POLDER FMFs were < 0.2 and Phy-DL FMFs were > 0.6, resulting in large differences (< -0.4). Figure S13 shows Phy-DL-, POLDER-, MISR-, and MODIS-estimated FMF frequencies at three levels (low: FMF < 0.5, medium: 0.5 < FMF < 0.8, high: FMF > 0.8) from 2008 to 2013. In the low-level category, MODIS and POLDER FMFs were more frequent than AERONET FMFs (50% and 20%, respectively), especially over the Amazon and western USA. The frequencies of MISR, Phy-DL, and AERONET FMFs in this category were in good agreement. In the medium-level category, high frequencies of Phy-DL and AERONET FMFs occurred over Australia and the Amazon (> 80%), and low frequencies of Phy-DL and AERONET FMFs occurred in sub-Saharan Africa, Central Africa, and Eastern China (< 30%). The MISR slightly overestimated the frequency of medium-level FMFs in Central Africa and underestimated it in Northern Australia and the Amazon. The frequencies of medium-level POLDER FMFs were underestimated over the Amazon and western USA and overestimated over Southeast China. MODIS was unable to capture medium-level FMFs globally, with frequencies of < 20%. High-level FMFs mainly appeared over areas experiencing biomass burning and urban regions, with frequencies commonly < 50%. The frequencies of MODIS, Phy-DL, and AERONET FMFs in the high-level category over Central Africa, Southern China, and the eastern USA agreed well. However, the frequencies of high-level MODIS FMFs were overestimated over the Amazon and underestimated over Northern India. The frequencies of high-level POLDER FMFs were captured well over Central Africa, but significantly underestimated over Northern India, Southern China, and the eastern USA. Moreover, MODIS was unable to capture high-level FMFs globally with frequencies of < 20%.

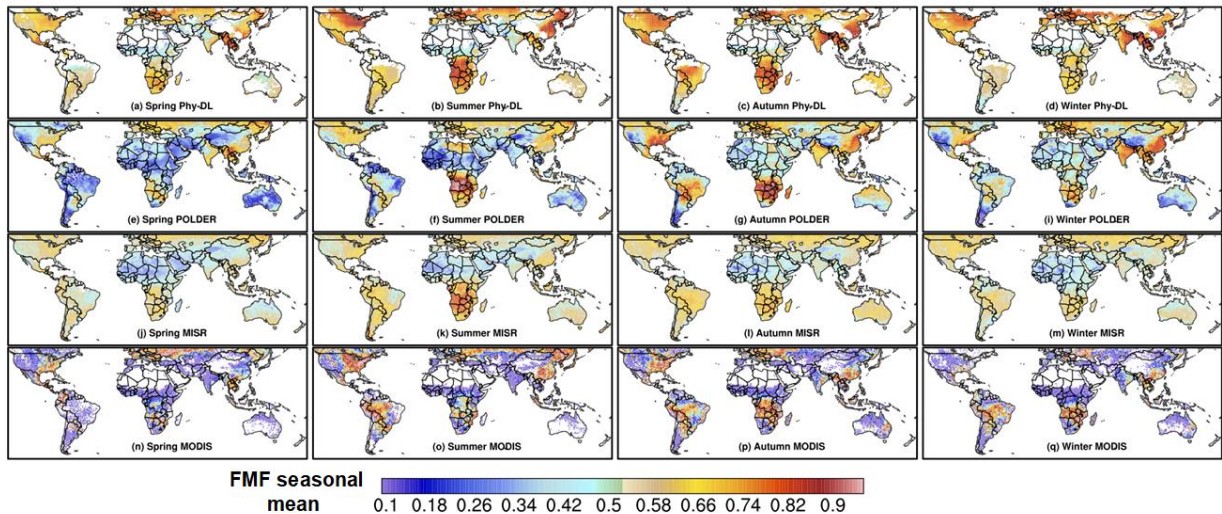

**FMF seasonal mean**

0.1 0.18 0.26 0.34 0.42 0.5 0.58 0.66 0.74 0.82 0.9

**Figure 13:** Seasonal mean FMF, averaged from 2008 to 2013, based on (from top to bottom) Phy-DL, POLDER, MISR, and MODIS. Columns from left to right are for spring, summer, autumn, and winter.

## 4. Data availability

The global land FMF dataset (2001–2020) developed in this study, Phy-DL FMF, is available at https://doi.org/10.5281/zenodo.5105617. The FMF data are in the Geotiff format on a daily scale.

## 5. Conclusion

Given the general lack of, or the poor quality of aerosol fine-mode fraction (FMF) over land, an improved long-term global aerosol FMF (at 500 nm) dataset (2001–2020) is developed over land with a hybrid retrieval algorithm combining physical and deep learning approaches called Phy-DL FMF. It was extensively evaluated against AERONET FMF retrievals, revealing its higher accuracy (RMSE = 0.136, based on 361,089 validation samples; 79.15% of the data fell within the ±20 % EE envelope) and generally good agreement with AERONET FMF with respect to its values, trends, and frequencies. In addition, independent validation was conducted based on SURFRAD FMF and the results showed the RMSE of Phy-DL FMF is 0.144 with 72.50% of the data fell within the ±20 % EE envelope.

Compared with physical–based (calculated using LUT-SDA, i.e., Phy-based FMF) and deep learning-based (DL-based FMF) FMF results, the accuracy of Phy-DL FMFs was substantially improved over five land types (forests, grasslands, croplands, urban area, and barren land), lessening the common problem of underestimation for high FMF values and

overestimations for low FMF values. Geographically, Phy-DL FMF captured the low FMFs well over the Saharan region, Central Asia, Australia, and southern South America, while Phy-based FMF showed significant overestimations. Phy-DL FMFs were also compared with three satellite-based official global FMF products (MISR, POLDER, and MODIS DT-based FMFs) using both AERONET FMF and SURFRAD FMF as references. Phy-DL FMF showed a significant improvement in terms of the accuracy and spatial distribution of trends. In Central Africa, Eastern China, Australia, the Amazon, and the western USA, Phy-DL FMFs agreed well with AERONET FMFs, while the other three satellite-based FMFs showed significant underestimations. In particular, in Southern Africa, the accuracy of the annual average was substantially improved, and the linear trends of Phy-DL FMF corresponded better with AERONET FMF. The Phy-DL FMF dataset also captured the seasonality and frequencies of FMFs well, thereby showing better agreement with AERONET FMFs.

By examining Phy-DL FMFs from 2001 to 2020, we found a general decreasing trend of $-1.9 \times 10^{-3}$ $yr^{-1}$ around the globe at the significance level of 95%, which was not revealed by AERONET point-scale measurements. However, both Phy-DL and AERONET FMFs showed significant increasing trends in FMF over the western USA and India ($> +3 \times 10^{-3}$ $yr^{-1}$). The new dataset captured high-level FMFs ($> 0.80$) over Southern China, South Asia, Eastern Europe, and the eastern USA. FMFs were consistently $< 0.3$ in Northwest China, the Saharan region, and southern South America, indicating coarse-particle desert emissions. The findings of various evaluations, especially the attempted explanations of the spatial-temporal variations and long-term trend changes, suggest that this newly developed dataset is sound, more accurate and thus useful for investigating the impact of fine- and coarse-mode aerosols on the atmospheric environment and climate, especially in gaining a deeper insight of fine-mode aerosols.

**Author contribution.**

**Xing Yan:** Writing - Original Draft, Conceptualization, Methodology, Supervision, Funding acquisition. **Zhou Zang:** Writing - Original Draft, Data Curation, Investigation, Visualization, Validation. **Zhanqing Li:** Writing - Review & Editing, Supervision, Funding acquisition. **Nana Luo**: Methodology, Writing - Original Draft, Investigation. **Chen Zuo:** Data Curation. **Yize Jiang:** Methodology, Software. **Dan Li:** Data Curation. **Yushan Guo:** Data Curation. **Wenji Zhao**: Writing - Review & Editing.  **Wenzhong Shi:** Writing - Review & Editing. **Maureen Cribb**: Writing - Review & Editing.

**Competing interests.**

The authors declare that they have no conflict of interest.

**Acknowledgements.**

The authors gratefully acknowledge the European Centre for Medium Range Weather Forecasts, MODIS, MISR, POLDER, SURFRAD and AERONET teams for their effort in making the data available.

**Financial support.**

This research is supported by the Natural Science Foundation of Beijing (8222058 and 8224088), the National Natural Science Foundation of China (42030606 and 91837204), the National Key Research and Development Plan of China (2017YFC1501702), and the Fundamental Research Funds for the Central Universities.

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
