# Peer review of "A global land aerosol fine-mode fraction dataset (2001–2020) retrieved from MODIS using hybrid physical and deep learning approaches"

_Earth System Science Data, 2021_

## Author Comment (AC1)

**Responses to RC1:**

This paper constructed a global aerosol fine mode fraction (FMF) dataset using combined physical and statistical methods. The dataset showed overall superior performance over existing satellite products. This is a good paper, providing a useful set of information to study the distribution and potentially climate impact of anthropogenic aerosols. I recommend publication of the paper after addressing a few questions. Although my recommendation is a major revision, the comments should not be difficulty to fix, but I think will increase of the credibility of the method and data.

**Response**: Thanks for your careful reading and comments. Your comments are valuable in improving the quality of our manuscript.

1. A 10 fold cross validation seems too simple. It would be interesting to see how the method perform on different sites or different years. Therefore, an out of site validation (e.g., predicting FMF at AERONET sites whose data are not used for training) or out of period validation (i.e., using part of the time series as training and the rest as validation) is highly recommended to increase the robustness of the results.

**Response**: Thank you for this valuable recommendation. We have conducted a validation based on measurements from four independent National Oceanic and Atmospheric Administration Surface Radiation Budget (SURFRAD) network sites not used for training in the deep-learning model. The SURFRAD network provides long-term, multi-band AOD observations at a temporal resolution of three minutes. As shown in **Figure R1**, the four sites (black triangles) are located across the US, covering different land types from forested land to barren land (**Tables R1 and R2**). The land types were based on MODIS MCD12C1 data from the International Geosphere-Biosphere Programme scheme.

[Figure]

**Figure R1**. (a) Locations of AERONET sites (green dots) and four independent SURFRAD sites (black triangles, **Table R1**) for the independent validation of the Phy-

DL FMF algorithm. The base map shows the land types from MODIS MCD12C1 data (the International Geosphere-Biosphere Programme scheme) in 2010. **Table R2** provides details about the land-type legend.

**Table R1**. SURFRAD sites used for out-of-site validation and their locations and land types.

| Sites | Longitude (°W) | Latitude (°N) | Land type |
|---|---|---|---|
| Desert Rock (DRA) | 116.02 | 36.62 | Barren or sparse |
| Fort Peck (FPK) | 105.10 | 48.31 | Grasslands |
| Goodwin Creek (GWN) | 89.87 | 34.25 | Woody savannas |
| Penn State (PSU) | 77.93 | 40.72 | Mixed forests |

**Table R2**. Land types and corresponding values from MODIS MCD12C1 data (the International Geosphere-Biosphere Programme scheme).

| Value | Land type | Value | Land type |
|---|---|---|---|
| 1 | Evergreen needleleaf | 9 | Savannas |
| 2 | Evergreen broadleaf | 10 | Grasslands |
| 3 | Deciduous needleleaf | 11 | Permanent wetlands |
| 4 | Deciduous broadleaf | 12 | Croplands |
| 5 | Mixed forests | 13 | Urban and built up |
| 6 | Closed shrubland | 14 | Crop natural vegetation mosaic |
| 7 | Open shrublands | 15 | Snow and ice |
| 8 | Woody savannas | 16 | Barren or sparse |

We used the same AERONET method (SDA) to calculate the FMF at the SURFRAD sites. SURFRAD data were not included in the model training, so SURFRAD FMFs can be regarded as the out-of-site validation for the Phy-DL FMF algorithm. **Figure R2** shows how SURFRAD and Phy-DL FMFs compare. The correlation coefficient (R) was 0.51, and the root-mean-square error (RMSE) was 0.144, somewhat poorer performance than AERONET validation results (i.e., R=0.68 and RMSE=0.136).

[Figure]

**Figure R2**. Phy-DL FMF at 500 nm as a function of SURFRAD FMF. The black and red solid lines are the 1:1 line and the best-fit line obtained from linear regression, respectively. The black dashed and dotted lines represent the expected error (EE) envelopes of ±20% and ±40%, respectively.

Furthermore, the Phy-DL FMF performance was validated at each SURFRAD site. **Figure R3** shows the bias of Phy-DL FMF (Phy-DL FMF minus SURFRAD FMF), retrievals falling within the ±20% EE envelope, and RMSEs at each SURFRAD site. In general, most of the sites have a mean bias and an RMSE lower than 0.1 and 0.15, respectively, with over 70% of the retrievals falling within the ±20% EE envelope. The out-of-site validation reveals that the Phy-DL FMF algorithm is reliable even in regions without AERONET sites for model training.

[Figure]

**Figure R3**. Boxplots of bias (Phy-DL FMF minus SURFRAD FMF), percentage of FMF estimates falling within the EE envelope of ±20% (dash-dotted line), and RMSEs at the four independent SURFRAD sites. The upper, middle, and lower lines in each box present the 75th, median, and 25th percentiles, respectively. The red point in each box represents the mean value of the FMF bias.

**Changes in the manuscript**: We have revised the manuscript as follows:
(1) Added a new figure (i.e., **Figure 4)** to the manuscript.

[Figure]

**Figure 4**. (a) The locations of AERONET sites (green points) and four independent SURFRAD sites (black triangles) for the independent validation of the Phy-DL FMF algorithm. The base map shows the land types from MODIS MCD12C1 data (the International Geosphere-Biosphere Programme scheme) in 2010. Table S1 provides details about the land-type legend. (b) Phy-DL FMF at 500 nm as a function of SURFRAD FMF. The black and red solid lines are the 1:1 line and the best-fit line obtained from linear regression, respectively. The black dashed and dotted lines represent the expected error (EE) envelopes of ±20% and ±40%, respectively. (c) Boxplots of bias (Phy-DL FMF minus SURFRAD FMF), percentage of FMF estimates falling within the EE envelope of ±20% (dash-dotted lines), and RMSEs at the four independent SURFRAD sites. The upper, middle, and lower lines in each box present the 75th, median, and 25th percentiles, respectively. The red point in each box represents the mean value of the FMF bias.

(2) Added **Table S3** and **Table S4** to the supplementary file.

**Table S3**. SURFRAD sites used for out-of-site validation and their locations and land types.

| Sites | Longitude (°W) | Latitude (°N) | Land type |
|---|---|---|---|
| Desert Rock (DRA) | 116.02 | 36.62 | Barren or sparse |
| Fort Peck (FPK) | 105.10 | 48.31 | Grasslands |
| Goodwin Creek (GWN) | 89.87 | 34.25 | Woody savannas |
| Penn State (PSU) | 77.93 | 40.72 | Mixed forests |

**Table S4**. Land types and corresponded values from MODIS MCD12C1 data (the International Geosphere-Biosphere Programme scheme).

| Value | Land type | Value | Land type |
|---|---|---|---|
| 1 | Evergreen needleleaf | 9 | Savannas |
| 2 | Evergreen broadleaf | 10 | Grasslands |
| 3 | Deciduous needleleaf | 11 | Permanent wetlands |
| 4 | Deciduous broadleaf | 12 | Croplands |
| 5 | Mixed forests | 13 | Urban and built up |
| 6 | Closed shrubland | 14 | Crop natural vegetation mosaic |
| 7 | Open shrublands | 15 | Snow and ice |
| 8 | Woody savannas | 16 | Barren or sparse |

(3) In section 3.1 entitled "Phy-DL FMF validation", we have added the following text:

Four sites from the National Oceanic and Atmospheric Administration's Surface Radiation Budget (SURFRAD) network were selected for the independent validation of the Phy-DL FMF algorithm, providing long-term, multi-bands AOD observations at the temporal resolution of three minutes. As shown in Figure 4a, the four sites (black triangles) are located across the US, covering different land types from forested land to barren land. The land types were based on MODIS MCD12C1 data from the International Geosphere-Biosphere Programme scheme. We used the same AERONET method (SDA) to calculate the FMF at the SURFRAD sites. SURFRAD data were not included in the model training, so SURFRAD FMFs can be used to do the out-of-site validation for the Phy-DL FMF algorithm. Figure 4b shows how SURFRAD and Phy-DL FMFs compare. The correlation coefficient (R) was 0.51, and the root-mean-square error (RMSE) was 0.144, somewhat different than AERONET validation results (i.e., R=0.68 and RMSE=0.136).

Furthermore, the Phy-DL FMF performance was validated at each SURFRAD site. Figure 4c shows the bias of Phy-DL FMF (Phy-DL FMF minus SURFRAD FMF), percentage of retrievals falling within the ±20% EE envelope, and RMSEs at each site. In general, most of the sites have a mean bias and an RMSE lower than 0.1 and 0.15, respectively, with over 70% of the retrievals falling within the ±20% EE envelope. The out-of-site validation reveals that the Phy-DL FMF algorithm is reliable even in regions without AERONET sites for model training.

2. How are the input variables selected? Are they all necessary or there might be some redundant inputs? Since MODIS AE is used as input, I don't see how some meteorology variables, such as boundary layer height, winds, RH are physically related to FMF.

**Response:** Thank you for the question. One of the most important reasons for considering meteorological variables in the deep-learning model is that the physical approach (SDA) does not include these potential meteorological influences. O'Neill et al. (2008) have reported that when the temperature is low, the error of the fine-mode AOD calculated by the SDA is clearly large (**Figure R4**). Although the developers of the SDA know about this issue, the relationship between meteorological factors and the FMF is complex, difficult to describe in equations. Therefore, we incorporate meteorological variables into deep learning to model these complex relationships with the FMF.

[Figure]

**Figure R4**. Variation in $\Delta\tau_f$ with detector temperature (PEARL CIMEL, May 1 to August 31, 2007). Copied from the Spectral Deconvolution Algorithm (SDA) technical memo.

We implemented the Generalized Additive Model (GAM) to fit the meteorological variables to the FMF and investigated their nonlinear relationships. **Figure R5** reveals that the meteorological variables considered, i.e., PBLH, temperature, surface pressure, RH, and wind speed, all had significant non-linear relationships with the FMF (at the 99% significance level). Both PBLH and surface pressure had similar influences on the FMF, i.e., a positive (negative) response when PBLH and surface pressure values were low (high). This is because high PBLH and surface pressure values can increase the diffusion of fine particles, decreasing the magnitude of the FMF (Tai et al., 2010). Meanwhile, the negative response of the FMF to wind speed also reflects the influence of fine-particle diffusion, as well as the contribution of dust particles strengthened by

wind speed (Luo et al., 2016). Increasing temperatures corresponded to decreasing FMFs, partly due to unfavorable diffusion conditions (Tai et al., 2010). On the other hand, more fine particles are released by heating during colder seasons than during warmer seasons (Ramachandran, 2007). RH had a strong positive influence on $PM_{2.5}$ concentrations when RH was between 25% and 75%. This reflects the secondary particle formation boosted by the increasing RH, contributing to the fine particles (Tai et al., 2010).

[Figure]

**Figure R5**. GAM fitting plots for the meteorological variables and the FMF. Shaded areas in the GAM plots indicate 95% confidence intervals, and the y-axes show the covariate and effective degrees of freedom of the smoothing. Asterisks (**) after each p-value indicate the 99% confidence interval of fitting.

Changes in the manuscript: We have revised the manuscript as follows:
(1) Added a new figure (i.e., **Figure S2)** in the supplementary file.

[Figure]

**Figure S2**. GAM fitting plots for the meteorological variables and the FMF. Shaded areas in the GAM plots indicate 95% confidence intervals, and the y-axes shows the

covariate and effective degrees of freedom of the smoothing. The asterisks (**) after each p-value indicate the 99% confidence interval of fitting.

(2) In section 2.3 entitled "Meteorological data", we have added the following text:

Previous studies have reported that meteorological variables are significantly correlated to fine-mode and coarse-mode aerosols. Tai et al. (2010), Liang et al. (2016), and Shen et al. (2018) all revealed that meteorological variables like temperature, RH, and wind speed explain much of the variations in $PM_{2.5}$ concentrations (> 50%). Xiang et al. (2019) and Gui et al. (2019) found a negative association between BLH and $PM_{2.5}$, and Kang et al. (2014) found that fine-mode aerosols and air pressure were correlated significantly. In this study, to investigate the correlation between meteorological variables and the FMF, we implemented the Generalized Additive Model (GAM). Figure S2 reveals that the meteorological variables considered in this study, i.e., PBLH, temperature, surface pressure, RH, and wind speed, all had significant non-linear relationships with the FMF (at the 99% significance level). Both PBLH and surface pressure had similar influences on the FMF, i.e., a positive (negative) response when PBLH and surface pressure values were low (high). This is because high PBLH and surface pressure values can increase the diffusion of fine particles, decreasing the magnitude of the FMF (Tai et al., 2010). Meanwhile, the negative response of the FMF to wind speed also reflects the influence of fine-particle diffusion, as well as the contribution of dust particles strengthened by wind speed (Luo et al., 2016). Increasing temperatures corresponded to decreasing FMFs, partly due to unfavorable diffusion conditions (Tai et al., 2010). On the other hand, more fine particles are released by heating during colder seasons than during warmer seasons (Ramachandran, 2007). RH had a strong positive influence on $PM_{2.5}$ concentrations when RH was between 25% and 75%. This reflects the secondary particle formation boosted by the increasing RH that contributed to the fine particles (Tai et al., 2010). Therefore, in this study, we used surface temperature, air pressure, PBLH, RH, and wind speed as inputs to the deep-learning model.

3. Although the proposed model produces the best FMF among different satellite products, there is still an obvious bias compared with AERONET. Also at some sites the agreement is low, such as Australia, North Africa, Mid East, etc. These all seem to be desert regions, could this be related to surface, or optical property assumption for non-spherical particles? Some more in-depth error analysis would be helpful to fully evaluate the performance of the technique.

**Response:** Thank you for the question. We analyzed this issue further. As shown in **Figure R6,** more than 75% of the sites located on barren land have low percentages of Phy-DL FMFs (< 60%) falling within the ±20% EE envelope. About 4% of the sites have high percentages of Phy-DL FMFs (> 90%) falling within the ±20% EE envelope.

[Figure]

**Figure R6**. Bar plots of the percentage of sites with > 90% of retrievals falling within the ±20% EE envelope (blue bars) and the percentage of sites with < 60% of retrievals falling within the ±20% EE envelope (red bars) for five land types.

The barren land type is a bright surface compared to other land types where the other sites are located (**Table R1**). AODs over the bright surface used for the Phy-DL FMF retrieval were significantly overestimated, with the worst performance compared to other vegetated land-cover types (Levy et al., 2010; Petrenko and Ichoku, 2013). This suggests that the performance of the Phy-DL FMF algorithm is poor when applied to regions with barren land. **Figure R7** shows the bias of the Phy-DL FMF and the percentage of retrievals falling within the EE envelope of ±20% as a function of NDVI. As NDVI increased from < 0.1 to > 0.8, the percentage of FMF retrievals falling within the ±20% EE envelope also rose from < 70% to > 85%, and the range of bias decreased significantly. Because a higher NDVI value indicates a darker surface, **Figure R7** reveals that the Phy-DL FMF algorithm performs better over dark surfaces than over bright surfaces, resulting in a lower accuracy over the barren land type than vegetated land types.

[Figure]

**Figure R7**. Box plots of the FMF bias (estimated FMF minus AERONET FMF) as a function of NDVI. The black horizontal dashed line indicates the zero bias. The gray dot in each box represents the mean value of the FMF bias. The upper, middle, and lower horizontal lines in each box show the 75th, median, and 25th percentiles, respectively. The green dots connected by the dashed curve are percentages of FMF retrievals falling within the EE envelope of ±20%.

The Ångström Exponent (AE) from the MODIS DT aerosol product is still highly

uncertain. The core of the SDA method relies on AE as input (Yan et al., 2017). The low accuracy of AE can significantly influence the performance of the Phy-DL FMF algorithm. As shown in **Figure R8**, AEs from the MODIS MOD08 product used as input to the Phy-DL FMF algorithm performed the worst over barren land, with the highest RMSE (> 1) and the lowest percentage of retrievals falling within the EE envelope of ±0.45 (< 45%). This would result in a lower performance of the Phy-DL FMF algorithm when applied to regions with barren land.

[Figure]

**Figure R8**. RMSEs (bars) and percentages of MOD08 AE falling within the EE envelope of ±0.45 (dash-dotted line) against AERONET observations for five land types. The EE envelope (±0.45) was adopted from Levy et al. (2013).

Changes in the manuscript: We have revised the manuscript as follows:
(1) Added new figures (i.e., **Figure S3** and **Figure S4**) to the supplementary file.

[Figure]

**Figure S3**. (a) Bar plots of the percentage of sites with > 90% of retrievals falling within the ±20% EE envelope (blue bars) and the percentage of sites with < 60% of retrievals falling within the ±20% EE envelope (red bars) for five land types. (b) Box plots of the FMF bias (estimated FMF minus AERONET FMF) as a function of NDVI. The black horizontal dashed line indicates the zero bias. The gray dot in each box represents the mean value of the FMF bias. The upper, middle, and lower horizontal lines in each box show the 75th, median, and 25th percentiles, respectively. The green dots connected by the dashed curve are percentages of FMF retrievals falling within the EE envelope of ±20%.

[Figure]

**Figure S4**. RMSEs (bars) and percentages of MOD08 AE falling within the EE envelope of ±0.45 (dash-dotted line) against AERONET observation for five land types. The EE envelope (±0.45) was adopted from Levy et al. (2013).

(2) In section 3.1 entitled "Phy-DL FMF validation", we have added the following text:

Figure S3a shows that more than 75% of the sites located on barren land have low percentages of Phy-DL FMFs (< 60%) falling within the EE envelope of ±20%. About 4% of the sites have high percentages of Phy-DL FMFs (> 90%) falling within the ±20% EE envelope. This suggests that the accuracy of Phy-DL FMF over barren land is much lower than over other land types. The barren land type is a bright surface compared to other land types where the other sites are located (Table S3). AODs over the bright surface used for the Phy-DL FMF retrieval were significantly overestimated, with the worst performance compared to other vegetated land-cover types (Levy et al., 2010; Petrenko and Ichoku, 2013). This suggests that the performance of the Phy-DL FMF algorithm is poor when applied to regions with barren land. Figure S3b shows the bias of the Phy-DL FMF and the percentage of retrievals falling within the EE envelope of ±20% as a function of NDVI. As NDVI increased from < 0.1 to > 0.8, the percentage of FMF retrievals falling within the ±20% EE envelope also rose from < 70% to > 85%, and the range of bias decreased significantly. The Ångström Exponent (AE) from the MODIS DT aerosol product is still highly uncertain. The core of the SDA method relies on AE as input (Yan et al., 2017). The low accuracy of AE can significantly influence the performance of the Phy-DL FMF algorithm. As shown in Figure S4, AEs from the

MODIS MOD08 product used as input to the Phy-DL FMF algorithm performed the worst over barren land, with the highest RMSE (> 1) and the lowest percentage of retrievals falling within the EE envelope of ±0.45 (< 45%). This would result in a lower performance of the Phy-DL FMF algorithm when applied to regions with barren land.

**References:**

Gui, K., Che, H., Wang, Y., Wang, H., Zhang, L., Zhao, H., Zheng, Y., Sun, T., and Zhang, X.: Satellite-derived $PM_{2.5}$ concentration trends over Eastern China from 1998 to 2016: relationships to emissions and meteorological parameters, Environmental Pollution, 247, 1125–1133, https://doi.org/10.1016/j.envpol.2019.01.056, 2019.

Kang, P., Feng, N., Wang, Z., Guo, Y., Wang, Z., Chen, Y., Zhan, J., Zhan, F. B., and Hong, S.: Statistical properties of aerosols and meteorological factors in Southwest China, Journal of Geophysical Research: Atmospheres, 119, 9914–9930, https://doi.org/10.1002/2014JD022083, 2014.

Levy, R. C., Remer, L. A., Kleidman, R. G., Mattoo, S., Ichoku, C., Kahn, R., and Eck, T. F.: Global evaluation of the Collection 5 MODIS dark-target aerosol products over land, Atmospheric Chemistry and Physics, 10, 10,399–10,420, https://doi.org/10.5194/acp-10-10399-2010, 2010.

Levy, R. C., Mattoo, S., Munchak, L. A., Remer, L. A., Sayer, A. M., Patadia, F., and Hsu, N. C.: The Collection 6 MODIS aerosol products over land and ocean, Atmospheric Measeurement Techniques, 6, 2989–3034, https://doi.org/10.5194/amt-6-2989-2013, 2013.

Liang, X., Li, S., Zhang, S., Huang, H., and Chen, S. X.: $PM_{2.5}$ data reliability, consistency, and air quality assessment in five Chinese cities, Journal of Geophysical Research: Atmospheres, 121, 10,220–10,236, https://doi.org/10.1002/2016JD024877, 2016.

Luo, N., An, L., Nara, A., Yan, X., and Zhao, W.: GIS-based multielement source analysis of dustfall in Beijing: a study of 40 major and trace elements, Chemosphere, 152, 123–131, 2016.

O'Neill, N., Eck, T., Smirnov, A., and Holben, B. (2008). Spectral deconvolution algorithm (SDA) technical memo.

Petrenko, M. and Ichoku, C.: Coherent uncertainty analysis of aerosol measurements from multiple satellite sensors, Atmospheric Chemistry And Physics, 13, 6777–6805, https://doi.org/10.5194/acp-13-6777-2013, 2013.

Ramachandran, S.: Aerosol optical depth and fine mode fraction variations deduced from Moderate Resolution Imaging Spectroradiometer (MODIS) over four urban areas in India, Journal of Geophysical Research: Atmospheres, 112, https://doi.org/10.1029/2007jd008500, 2007.

Shen, L., Jacob, D. J., Mickley, L. J., Wang, Y., and Zhang, Q.: Insignificant effect of climate change on winter haze pollution in Beijing, Atmospheric Chemistry and Physics, 18, 17,489–17,496, https://doi.org/10.5194/acp-18-17489-2018, 2018.

Tai, A. P. K., Mickley, L. J., and Jacob, D. J.: Correlations between fine particulate matter (PM$_{2.5}$) and meteorological variables in the United States: implications for the sensitivity of PM$_{2.5}$ to climate change, Atmospheric Environment, 44, 3976–3984, https://doi.org/10.1016/j.atmosenv.2010.06.060, 2010.

Xiang, Y., Zhang, T., Liu, J., Lv, L., Dong, Y., and Chen, Z.: Atmosphere boundary layer height and its effect on air pollutants in Beijing during winter heavy pollution, Atmospheric Research, 215, 305–316, https://doi.org/10.1016/j.atmosres.2018.09.014, 2019.

Yan, X., Li, Z., Shi, W., Luo, N., Wu, T., and Zhao, W.: An improved algorithm for retrieving the fine-mode fraction of aerosol optical thickness, part 1: Algorithm development, Remote Sensing of Environment, 192, 87-97, 10.1016/j.rse.2017.02.005, 2017.

---

## Author Comment (AC2)

**Responses to RC2:**

The authors need to address the following major points.
**Response**: We thank the reviewer for insightful and very pertinent comments to improve the paper.

1. The motivation of this manuscript is to address the following issues raised by the authors, the first issue is the Chen et al (2020) method is not applied on a global scale, the second issue is that 'Zhang et al (2016) noted that satellite –measured multi-spectral reflectance of ground-based data alone was not sufficient to retrieve FMFs with high accuracy' (the meaning of this sentence is not quite clear to me), my interpretation is that the use of (only) spectral information from satellite measurements is not enough for the retrieval of FMF, this two reasons are not really solid enough to continue the work proposed in this manuscript. Both manuscripts mentioned above focus on the original level 2 spatial scale (with a very quick look at these two papers), while this work focuses on the level 3 data, the average of spatial resolution from kilometre to degree can make all problems significantly easier, both technically and scientifically. Even later, the authors point to their own previous publication (Yan et al., 2021b) and claimed 'As shown by Yan et al. (2021b), the global land Phy-based FMF is still unreliable.', in which the authors even started 'seasonal FMF characteristics and trends' analysis using the 'unreliable' dataset. This is quite misleading for the understanding of the motivation for the developments in this manuscript, even from the very first step. I think the work should start from level 2 dataset rather than level 3.

**Response**: Thank you for this question. The key motivation for our paper is to improve the MODIS-based global land FMF retrieval accuracy. The MODIS-based global aerosol product MOD08/MYD08 does not include FMF data because of its high uncertainty. In this study, we only use MODIS level 3 data (i.e., AOD and AE from the MOD08 product) because it is more accurate than level 2 data (Kharol et al., 2011).

We mention the issue "Zhang et al. (2016) noted that satellite-measured multi-spectral reflectance of ground-based data alone was not sufficient to retrieve FMFs with high accuracy" because O'Neill et al. (2008) showed that when the temperature is low, the error of the fine-mode AOD calculated by the physical method, i.e., the Spectral Deconvolution Algorithm (SDA), is clearly large (SDA technical memo, O'Neill et al., 2008, **Figure R1**). So if we want to obtain more accurate FMF retrievals, meteorological variables must be considered. However, the relationship between meteorological factors and FMF is complex, difficult to describe by equations. Therefore, we used a deep-learning model to consider the impact of meteorological factors. This is another motivation of this research. We mention the issue "As shown by Yan et al. (2021b), the global land Phy-based FMF is still unreliable." We note that Yan et al. (2021b) still used the SDA to calculate the FMF without considering the impact of meteorological factors.

[Figure]

**Figure R1**. Variation in $\Delta\tau_f$ with detector temperature (PEARL CIMEL, May 1 to August 31, 2007). Copied from the Spectral Deconvolution Algorithm (SDA) Technical memo.

**Changes in the manuscript:** In the Introduction, we express the motivation more clearly as follows:

Zhang et al. (2016) noted that satellite-measured multi-spectral reflectance of ground-based data alone was not sufficient to retrieve FMFs with high accuracy. O'Neill et al. (2008) showed that when the temperature is low, the error of the fine-mode AOD calculated by the physical method, i.e., the Spectral Deconvolution Algorithm (SDA), is clearly large (SDA technical memo, O'Neill et al., 2008). Although this issue has long been known, the relationship between meteorological factors and FMF is complex, difficult to describe by equations in the SDA. Benefiting from its powerful ability to describe nonlinear relationships, using a deep-learning model may overcome the deficiencies of the SDA in calculating FMFs.

2. The second major point the authors highlighted is that the method is a combination of physical and deep learning approach, this is also quite confusion. As presented in section 2.4, this dataset is created by the new algorithm described in this manuscript for the first time, however, my feeling is that it is a mixture of previous publications without a clear description of the method itself. Meanwhile, the authors mark this paper as a 'Data description paper' in the submission. I never saw a 'Data description paper' without a clear and solid 'Method description paper' before. The key physical part is the LUT-SDA, described in equation (1). Firstly, I am confused why we have two af here? Second, I can not understand how the combination between the physical method and deep learning is achieved. To my personal view, it is still a deep learning method with certain parameters from some physical derivation, however, your major input, aerosol

optical thickness, is derived from a physical model (MODIS retrieval algorithm) as well, you cannot claim it is a combination of physical and deep learning method becuase some inputs for the deep learning is from a product derived from a physical method.

**Response**: There are three modes for the combined physical and machine-learning model shown in **Figure R2** (Prof. Shen Huanfeng, 2021 China Annual Conference on Theories and Methods of Geographic Information Science).

The first mode is called "Concatenation": The physical model outputs as the machine-learning model inputs or the machine-learning model outputs as the physical model inputs (**Figure R3a**), which is the exact mode employed in our study. Shen et al. (2018) also used this mode to estimate $PM_{2.5}$ by integrating the physical retrieval of AOD and other datasets via the deep-learning model.

The second mode is called "Embedding": This mode uses a machine-learning model as the simulator to replace one part of the physical model, accelerating the overall computational process (**Figure R3b**). For example, Krasnopolsky et al. (1998) used the neural network approach to approximate the atmospheric longwave radiation parameterization for the NCAR Community Atmospheric Model, resulting in faster estimation results.

The third mode is called "Integration": This involves adding physical constraints into the objective function to optimize the result by solving the minimum value of this objective function (**Figure R3c**). T. Li et al. (2021) used a geographically weighted loss function, which served as the spatial constraint for building their AOD–$PM_{2.5}$ relationship.

Therefore, our method using the LUT-SDA FMF as input for the deep-learning model is a way of combining physical and machine-learning models, i.e., concatenation.

[Figure]

**Figure R2.** Three combination modes for physical and machine-learning models (copied from the Microsoft PowerPoint presentation prepared by Prof. Shen Huanfeng

and presented at the 2021 China Annual Conference on Theories and Methods of Geographic Information Science).

**a. Concatenation:**

[Figure]

Chen et al. (2018)

**b. Embedding:**

[Figure]

**c. Integration:**

[Figure]

**Figure R3.** Core structures for the three combination modes (a. concatenation, b. embedding, c. integration) for physical and machine-learning models (copied from the Microsoft PowerPoint presentation prepared by Prof. Shen Huanfeng and presented at the 2021 China Annual Conference on Theories and Methods of Geographic Information Science).

In the revised paper, we have clarified Equation (1). We have also added detailed information about the Phy-DL FMF calculation in the main text and the Supplementary Information document. A new detailed schematic diagram was added to the Supplementary Information document (see Figure S1 below).

==Changes in the manuscript:==
We added a new figure (Figure 2) to section 2.4 entitled "Combining physical and deep-learning models (Phy-DL) for retrieving FMFs".

[Figure]

**Figure 2.** Visual representation of the SDA-based FMF retrieval LUT.

**More details in section 2.4**:

In this study, we used a "concatenation" mode to combine a physical model and a deep-learning model, i.e., the outputs of the physical model were used as the inputs for the deep-learning model (Figure S1). The physical model used is the LUT-SDA (Yan et al., 2017). The LUT-SDA is designed for satellite FMF retrievals when only AODs at two wavelengths are available (such as DT AOD products). As shown in Eq. (1) of the SDA (O'Neill et al., 2001a), a minimum of AODs at three wavelengths are needed to first obtain the Ångström exponent (AE) derivative ($\alpha'$). The AE of the fine-mode AOD ($\alpha_f$) and the FMF can then be calculated.

$$\begin{cases} \alpha_f = \dfrac{1}{2(1-a)} \{(\alpha-\alpha_c - \dfrac{\alpha'-\alpha_c'}{\alpha-\alpha_c} + b^*) + [(\alpha-\alpha_c - \dfrac{\alpha'-\alpha_c'}{\alpha-\alpha_c} + b^*)^2 + 4c^*(1-a)]^{1/2}\} + \alpha_c \\ \\ \text{FMF=} \dfrac{\alpha-\alpha_c}{\alpha_f - \alpha_c} \end{cases} \quad (1)$$

where a, b*, c*, $\alpha_c'$, and $\alpha_c$ are fixed parameters described in section 1 of the Supplementary Information document, based on O'Neill (2010). Since AODs at two wavelengths are not sufficient to calculate $\alpha'$, for the global physically based FMF retrieval, we first divide the whole world into nine regions [as done by Sayer et al. (2014)] and use historical AERONET observed data to determine $\alpha'$ value ranges in these regions. The $\alpha'$ range of values is based on the first and third quartiles of AERONET measurements in different seasons. For example, in Southeast Asia, $\alpha'$ ranges from 0.12 to 0.60 in spring (Yan et al., 2021). In these nine regions, a set of hypothetical values for $\alpha'$ [as determined by Yan et al. (2021)], $\alpha_f$, and AE ($\alpha$) are imported into the SDA [Eq. (1)] to build the relationship with FMF (Figure 2).

Different LUTs based on the SDA for these regions are thus created. Based on the constructed LUT, initial results are obtained using a cost function:

$$(FMF^1, \alpha'^1, \alpha_f^1) = \min[(LUT - SDA_{AE} - MODIS_{AE})]^2 , \quad (2)$$

where $FMF^1$, $\alpha'^1$, and $\alpha_f{}^1$ are uncorrected initial results of FMF, $\alpha'$, and $\alpha_f$ by the LUT-SDA, $LUT-SDA_{AE}$ is the $\alpha$ in the LUT, and $MODIS_{AE}$ is the MODIS MOD08 DT-based AE. After performing the $\alpha'$ bias error correction [described in Supplementary Information, Section 2, O'Neill et al. (2003)] and the mean of extreme (MOE) modification [described in Supplementary Information, Section 3, O'Neill et al. (2008)], the final FMF output is

$$FMF_{output} = \frac{\alpha - \alpha_c}{\alpha_{fcorrected}^1 - \alpha_c} \qquad . \qquad (3)$$

**We added detailed information about the Phy-DL FMF calculation in the Supplementary Information document**:

A new detailed flowchart was added to describe the Phy-DL FMF calculation:

[Figure]

**Figure S1**. Schematic diagram describing the Phy-DL FMF calculation in this study.

**1. The parameters in Eq. (1)**

The parameters in Eq. (1) are same as those described by O'Neill et al. (2010):

$$\alpha_f = \frac{1}{2(1-a)} \left\{ (\alpha - \alpha_c - \frac{\alpha' - \alpha_c{'}}{\alpha - \alpha_c} + b^*) + [(\alpha - \alpha_c - \frac{\alpha' - \alpha_c{'}}{\alpha - \alpha_c} + b^*)^2 + 4c^*(1-a)]^{1/2} \right\} + \alpha_c . \quad (1)$$

The parameters are:

$$\begin{cases} a = \left( a_{lower} + a_{upper} \right)/2 \\[2mm] a_{upper} = -0.22 \\[2mm] a_{lower} = -0.3 \end{cases}$$

$$\begin{cases} b^* = b + 2\alpha_c\, a \\[2mm] b = \left( b_{lower} + b_{upper} \right)/2 \\[2mm] b_{upper} = 10^{-0.2388}\lambda^{1.0275} \\[2mm] b_{lower} = 0.8 \end{cases}$$

where $\lambda$ is the reference wavelength (µm). In this study, $\lambda$ is 0.5 µm.

$$
\begin{cases}
c^* = c + (b + a\,\alpha_c)\alpha_c - \alpha_c' \\[2mm]
c = \left(c_{lower} + c_{upper}\right)/2 \\[2mm]
c_{upper} = 10^{0.2633}\,\lambda^{-0.4683} \\[2mm]
c_{lower} = 0.63
\end{cases}
$$

$$\alpha_c = -0.15 \quad \text{and} \quad \alpha_c' = 0$$

**2. $\alpha'$ bias error correction**

This study used Appendix A1 of O'Neill et al. (2003) to correct the $\alpha'$ bias and propagate this correction through all derived parameters:

$$\alpha'_{error} = 0.65 \times exp[-(FMF^1 - 0.78)^2 / (2 \times 0.18^2)],$$

where $FMF^1$ is the uncorrected estimate of $FMF$ as shown in Eq. (2) of the main paper. Then

$$\alpha'_{corrected} = \alpha'^1 + \alpha'_{error} \quad ,$$

$$t_{corrected} = \alpha - \alpha_c - \frac{\alpha'_{corrected} - \alpha_c'}{\alpha - \alpha_c} \quad ,$$

$$D_{corrected} = \sqrt{(t_{corrected} + b^*)^2 + 4(1 - a)\,c^*} \quad ,$$

$$\alpha_{f_{corrected}} = \frac{1}{2(1-a)}(t_{corrected} + b^* + D_{corrected}) + \alpha_c \quad ,$$

$$FMF_{corrected} = \frac{\alpha - \alpha_c}{\alpha_{f_{corrected}} - \alpha_c} \quad .$$

**3. Mean of extreme (MOE) modification**

The error in $\alpha_f$ derived by the SDA is (O'Neill et al., 2003):

$$
\Delta\alpha_f^2 = \left(k_1 \frac{\partial\alpha_f}{\partial\alpha'} + k_2 \frac{\partial\alpha_f}{\partial\alpha}\right)^2 \left(\frac{\Delta\tau_a}{\tau_a}\right)^2 + \left(\frac{\partial\alpha_f}{\partial a}\Delta a\right)^2 + \left(\frac{\partial\alpha_f}{\partial b}\Delta b\right)^2 + \left(\frac{\partial\alpha_f}{\partial c}\Delta c\right)^2
$$

$$
+ \left(\frac{\partial\alpha_f}{\partial\alpha'_c}\Delta\alpha'_c\right)^2 + \left(\frac{\partial\alpha_f}{\partial\alpha_c}\Delta\alpha_c\right)^2
$$

where $k_1 = 10$, $k_2 = -2.5$, $\Delta\tau_a$ is the nominal root mean square error in AOD at the reference wavelength, $\tau_a$ is the AOD at the reference wavelength (in this study, 0.5 μm), $\Delta\alpha'_c = 0.15$, $\Delta\alpha_c = 0.15$, and

$$\left\{ \begin{array}{l} \Delta a = (a_{upper} - a_{lower})/2 \\[2mm] \Delta b = (b_{upper} - b_{lower})/2 \\[2mm] \Delta c = (c_{upper} - c_{lower})/2 \, . \end{array} \right.$$

In $\Delta\alpha_f^{\,2}$,

$$\frac{\partial\alpha_f}{\partial\alpha'} = \frac{-1}{FMF_{corrected}\ D_{corrected}},$$

$$\frac{\partial\alpha_f}{\partial\alpha} = \frac{t_+}{FMF_{corrected}\ D_{corrected}},$$

$$t_+ = \alpha - \alpha_c - \frac{\alpha'_{corrected} - \alpha'_c}{\alpha - \alpha_c},$$

$$\frac{\partial\alpha_f}{\partial a} = \frac{(\alpha_{f_{corrected}} - \alpha_c)}{(1 - a)} + \frac{1}{D_{corrected}}\left(\alpha_c(2\alpha_{f_{corrected}} - \alpha_c) - \frac{c^*}{(1 - a)}\right),$$

$$\frac{\partial\alpha_f}{\partial b} = \frac{\alpha_{f_{corrected}}}{D_{corrected}},$$

$$\frac{\partial\alpha_f}{\partial c} = \frac{1}{D_{corrected}},$$

$$\frac{\partial\alpha_f}{\partial\alpha'_c} = \frac{1}{D_{corrected}}\left(\frac{1}{FMF_{corrected}} - 1\right),$$

$$\frac{\partial\alpha_f}{\partial\alpha_c} = \frac{t_{corrected}}{D_{corrected}}\left(\frac{1}{FMF_{corrected}} - 1\right).$$

When we obtain the $\Delta\alpha_f$ $(=\sqrt{\Delta\alpha_f^{\,2}})$, the SDA sets the theoretical maximum of $\alpha_f$ as:

$$\alpha_{fTMAX} = min(4, 10^{(0.18*log10(\lambda)+0.57)}).$$

Then,

$$\alpha_{fMAX} = \alpha_{f_{corrected}} + \Delta\alpha_f ,$$

$$\alpha_{fMin} = \alpha_{f_{corrected}} - \Delta\alpha_f .$$

If $\alpha_{fMAX} > \alpha_{fTMAX}$, $\alpha_{fMAX} = \alpha_{fTMAX}$.

If $\alpha_{fMin} > \alpha_{fTMAX}$, $\alpha_{fMin} = \alpha_{fTMAX}$.

The final output of corrected FMF ( $FMF_{\text{output}}$ ) is:

[Figure]

where $m = 8$ and $\Delta\alpha = k_2 \dfrac{\Delta\tau_a}{\tau_a}$.

3.The third major point is the comparison between different satellite products, the authors need to be aware that it is really the same parameter in the comparison or not, 'bad' agreements between satellite FMF products and the AERONET FMF product do not reveal anything because these FMF are not the same FMF due to different assumptions in particle size distribution and the 'cutting criteria' in the level 2 retrieval process, the FMF derived from this paper is somehow with knowledge (maybe certain inputs as well, not sure about it) from AERONET, it is not surprise at all to have a better agreement with AERONET measurements later. It really makes no sense to include the

MODIS comparison since it is already removed in the new version of dataset. The remove of FMF product in MODIS dataset also indicates how uncertain such a parameter can be.

**Response**: The validation for satellite FMF products followed Levy et al. (2007) (**Figure R4**). They pointed out that "although the actual products provided by MODIS and AERONET are not necessarily physically identical, in many cases they are comparable", supporting the direct comparison between satellite FMF products and AERONET FMFs. In addition, other validations of satellite FMF products, from POLDER (Wei et al., 2020; Li et al., 2020) to MISR (Dey and Di Girolamo, 2010), also directly used AERONET FMFs for comparison purposes. Zhang et al. (2021) inter-compared their newly retrieved FMFs and MODIS MYD04 FMFs directly with AERONET FMFs (**Figure R5**). Levy et al. (2007) made it clear that "The improvement to the MODIS FMF product is mainly its correlation to AERONET FMF." Therefore, we inter-compared satellite FMF products (from MISR, POLDER, and MODIS) and Phy-DL FMFs directly with AERONET FMFs.

[Figure]

**Figure R4**. MODIS aerosol size retrievals compared with AERONET derived products. The solid shapes and error bars represent the mean and standard deviation of the MODIS retrievals, in 20 bins of AERONET-derived product. Both the retrievals from V5.1 (orange) and V5.2 (green) are shown. The regressions (solid lines) are for the cloud of all points (not shown). The η over land retrieved at 0.55 mm, compared with AERONET η retrieved by the O'Neill et al. (2003) method. Note that η is defined differently for MODIS and AERONET and that we only show results for $\tau > 0.20$. Copied from Levy et al. (2007).

[Figure]

**Figure R5**. Comparison between the results of this study and MODIS FMFs with AERONET FMFs. Copied from Zhang et al. (2021).

Furthermore, we conducted a validation based on measurements from four independent National Oceanic and Atmospheric Administration Surface Radiation Budget (SURFRAD) network sites not used for training in the deep-learning model. The SURFRAD network provides long-term, multi-bands AOD observations at a temporal resolution of three minutes. As shown in **Figure R6**, the four sites (black triangles) are located across the US, covering different land types from forested land to barren land.

[Figure]

**Figure R6**. (a) Locations of AERONET sites (green dots) and four independent SURFRAD sites (black triangles, **Table R1**) for the independent validation of the Phy-DL FMF algorithm. The base map shows the land types from MODIS MCD12C1 data (the International Geosphere-Biosphere Programme scheme) in 2010. **Table R2** provides details about the land-type legend.

**Table R1**. The sites from SURFRAD used for out of site validation and their locations.

| Sites | Longitude (ºW) | Latitude (ºN) | Land type |
|-------|----------------|---------------|-----------|

| | | | |
|---|---|---|---|
| Desert Rock (DRA) | 116.02 | 36.62 | Barren or sparse |
| Fort Peck (FPK) | 105.10 | 48.31 | Grasslands |
| Goodwin Creek (GWN) | 89.87 | 34.25 | Woody savannas |
| Penn State (PSU) | 77.93 | 40.72 | Mixed forests |

**Table R2**. Land types and corresponding values from MODIS MCD12C1 data (the International Geosphere-Biosphere Programme scheme).

| Value | Land type | Value | Land type |
|---|---|---|---|
| 1 | Evergreen needleleaf | 9 | Savannas |
| 2 | Evergreen broadleaf | 10 | Grasslands |
| 3 | Deciduous needleleaf | 11 | Permanent wetlands |
| 4 | Deciduous broadleaf | 12 | Croplands |
| 5 | Mixed forests | 13 | Urban and built up |
| 6 | Closed shrubland | 14 | Crop natural vegetation mosaic |
| 7 | Open shrublands | 15 | Snow and ice |
| 8 | Woody savannas | 16 | Barren or sparse |

The inter-comparison results in **Figure R7** shows that when validated by independent FMF observations not used for training in the deep-learning model (SURFRAD FMFs), Phy-DL FMFs still outperform the other satellite products, with the highest R (0.51), lowest RMSE (0.143), and the greatest number of retrievals falling within the EE envelopes of ±20% (69.08%) and ±40% (89.05%). POLDER and MISR FMFs have the second best performance. PODLER results have an RMSE of 0.232 and R of 0.32, with 76.10% (48.23%) of retrievals falling within the EE envelope of ±40% (20%). MISR results have an R and RMSE of 0.22 and 0.212, respectively, with 82.61% (45.38%) of retrievals falling within the EE envelope of ±40% (20%). MODIS results were the poorest, with an especially high RMSE (0.465) and low percentages of retrievals falling within the EE envelopes of ±40% (37.23%) and 20% (18.09%). Overall, at the independent SURFRAD sites, Phy-DL FMFs are still more accurate and reliable than the other FMF products.

[Figure]

**Figure R7**. Evaluation of monthly mean (a) MISR (550 nm), (b) POLDER (490 nm), (c) MODIS (550 nm), and (d) Phy-DL FMFs (500 nm) against SURFRAD FMFs (500 nm) from 2008 to 2013. Black and red solid lines are 1:1 reference lines and best-fit lines from linear regression, respectively. Black dashed and dotted lines represent the EE envelopes of ±20% and ±40%, respectively. The number of samples (N), root-mean-square error (RMSE), correlation coefficient (R), and linear regression relation are given in each panel.

Changes in the manuscript:
**(1)** We have added this validation between independent SURFRAD FMFs and satellite products in the Supplementary Information document:

[Figure]

**Figure S7**. Evaluation of (a) MISR (550 nm), (b) POLDER (490 nm), (c) MODIS (550 nm), and (d) Phy-DL FMFs (500 nm) against SURFRAD FMFs (500 nm) from 2008 to 2013. Black and red solid lines are 1:1 reference lines and best-fit lines from linear regression, respectively. Black dashed and dotted lines represent the EE envelopes of ±20% and ±40%, respectively. The number of samples (N), root-mean-square error (RMSE), correlation coefficient (R), and linear regression relation are given in each panel.

**(2)** In section 3.4 entitled "Comparison with other satellite-based FMF products", we have added:

The inter-comparison results in **Figure R7** shows that when validated by independent FMF observations not used for training in the deep-learning model (SURFRAD FMFs), Phy-DL FMFs still outperform the other satellite products, with the highest R (0.51), lowest RMSE (0.143), and the greatest number of retrievals falling within the EE envelopes of ±20% (69.08%) and ±40% (89.05%). PODLER results have an RMSE of 0.232 and R of 0.32, with 76.10% (48.23%) of retrievals falling within the EE envelope of ±40% (20%). MISR results have an R and RMSE of 0.22 and 0.212, respectively, with 82.61% (45.38%) of retrievals falling within the EE envelope of ±40% (20%). MODIS results were the poorest, with an especially high RMSE (0.465) and low percentages of retrievals falling within the EE envelopes of ±40% (37.23%) and 20% (18.09%). Overall, at the independent SURFRAD sites, Phy-DL FMFs are still more accurate and reliable than the other FMF products.

4. The fourth major point is the application of this dataset, my personal view is that it

is too early to sell it as a dataset which is mutual enough for a trend analysis, especially since the application of (both fine and coarse as total) aerosol optical thickness is still quite questionable. Even at the last part of the manuscript, the authors claim some significant trends from the new satellite product are not revealed by AERONET measurements due to the scale issue, this simply reveals the limited representative of these AERONET sites in your regions, rather than anything with respect to the satellite data quality.

**Response**: Thank you for this question. There have been several studies on improving FMF retrievals, some including published datasets, e.g., Chen et al. (2020) (*https://pan.baidu.com/share/init?surl=PhHDLuXv1ltEPZN1wQ68lA,* password: *aero*) and Zhang et al. (2021) (can be requested from the corresponding author: *lizq@radi.ac.cn*). Chen et al. (2020) compared FMF validation studies with a focus on East Asia for different methods (Table R3). Compared with these previous retrieval results, Phy-DL FMF retrievals have better accuracy and lower uncertainty (R=0.68, RMSE=0.136, 90.53% of retrievals falling in the EE envelope of ±40%). Although not highly accurate, many studies have applied these published FMF datasets. Ramachandran (2007) used the MODIS FMF to explore FMF seasonal characteristics in India (**Figure R8**). B. Li et al. (2012) also used the MODIS FMF to investigate the spatial and temporal variations of global fine- and coarse-mode AODs (**Figure R9**). The MODIS global aerosol FMF (MOD08/MYD08) has been recalculated from MODIS regional aerosol products (MOD04/MYD04). Due to the high uncertainty of the MOD04/MYD04 FMF, the latest MODIS C6 global aerosol FMF is no longer available. However, Zhang et al. (2021) used the MODIS C6 MOD04/MYD04 FMF for comparison purposes with their newly retrieved FMF, both plotted as a function of AERONET FMF (**Figure R5**). In addition, Zhang and Li (2015) applied the MODIS C6 MOD04/MYD04 FMF for PM$_{2.5}$ retrievals to isolate the fine-particle contribution (**Figure R10**). Wei et al. (2021) used the FMF derived from POLDER to retrieve PM$_{10}$ over China. The key parameter, the columnar volume-to-extinction ratio VE$_{10}$, was retrieved by building relationships with FMF (**Figure R11**).

**Table R3**. Overview of published FMF validation studies with a focus on East Asia for different methods applied to data from multi-spectral sensors. Copied from Chen et al. (2020).

| Method | literature | Sensor | RMSE | % in EE[a] | Surface |
|---|---|---|---|---|---|
| Collection 6 | Levy et al., 2013 validated in Yan et al., 2017 (Table 1) | MODIS | 0.340 | 20% | Dark Target |
| LUT-SDA | Yan et al., 2017 (Table 1) | MODIS | 0.168 | 80% | Dark Target |
| NNAero | This paper (Fig. 10d) | MODIS | **0.157** | **91%** | **No Limit** |
| YAER | Choi et al., 2016 (Fig. 11a) | GOCI | 0.264 | —[b] | **No Limit** |

[a] EE envelopes is ± 0.4.
[b] "—" means not given in the literature.

[Figure]

**Figure R8**. Averaged intra-annual (2001–2005) fine-mode fraction values at (a) Chennai, (b) Mumbai, (c) Kolkata, and (d) New Delhi. Vertical bars denote ±1σ from the mean. Copied from Ramachandran (2007).

[Figure]

**Figure R9**. Global inter-annual variation of fine-mode (AOD$_f$), coarse-mode (AOD$_c$), and total AOD. Copied from Li et al. (2012).

[Figure]

**Figure R10**. Flowchart of the PM$_{2.5}$ remote sensing method. Copied from Zhang and Li (2015).

$$VE_{10} = 0.3178FMF^2 - 0.8199FMF + 0.69194 \ (0 < FMF \le 1) \qquad (16)$$

[Figure]

**Figure R11**. Relationship between the FMF and VE10 at seven typical AERONET sites. Colors represent the frequency of samples in each bin, with FMF intervals of 0.02 and VE$_{10}$ intervals of 0.02 μm$^3$ μm$^{-2}$. There are 1193 valid intervals in total. The black dots and whiskers represent the mean values and standard deviations for four aerosol types. The black solid line shows the fitting curve. Copied from Wei et al. (2021).

This study examined not only the accuracy of Phy-DL FMFs (AERONET validation), but the performances of FMF products from different satellite products using independently derived SURFRAD FMFs (i.e., SURFRAD FMFs not used in the model training). In the revised manuscript, the new SURFRAD validation shows that the Phy-DL FMF retrieval is the most accurate of three FMF products (**Figure R7**). Global trends in Phy-DL and AERONET FMFs over a twenty-year period agreed well. These results suggest that the Phy-DL FMF is reliable enough, even better than existing products. Considering the wide application of MISR and POLDER FMF products that are still highly uncertain, we believe it is reasonable to publish our Phy-DL FMF as a new product, given its demonstrated improved accuracy.

**References:**

Chen, X., de Leeuw, G., Arola, A., Liu, S., Liu, Y., Li, Z., and Zhang, K.: Joint retrieval of the aerosol fine mode fraction and optical depth using MODIS spectral reflectance over northern and eastern China: artificial neural network method, Remote Sensing of Environment, 249, https://doi.org/10.1016/j.rse.2020.112006, 2020.

Dey, S. and Di Girolamo, L.: A climatology of aerosol optical and microphysical properties over the Indian subcontinent from 9 years (2000–2008) of Multiangle Imaging Spectroradiometer (MISR) data, Journal of Geophysical Research: Atmospheres,115, https://doi.org/10.1029/2009JD013395, 2010.

Kharol, S. K., Badarinath, K. V. S., Sharma, A. R., Kaskaoutis, D. G., and Kambezidis, H. D.: Multiyear analysis of Terra/Aqua MODIS aerosol optical depth and ground observations over tropical urban region of Hyderabad, India, Atmospheric Environment, 45, 1532–1542, https://doi.org/10.1016/j.atmosenv.2010.12.047, 2011.

Krasnopolsky, V. M., Fox-Rabinovitz, M. S., and Chalikov, D. V.: New approach to calculation of atmospheric model physics: accurate and fast neural network emulation of longwave radiation in a climate model, Monthly Weather Review, 133, 1370–1383, https://doi.org/10.1175/mwr2923.1, 2005.

Levy, R. C., Remer, L. A., Mattoo, S., Vermote, E. F., and Kaufman, Y. J.: Second-generation operational algorithm: retrieval of aerosol properties over land from inversion of Moderate Resolution Imaging Spectroradiometer spectral reflectance, Journal of Geophysical Research: Atmospheres, 112, https://doi.org/10.1029/2006jd007811, 2007.

Li, B., Su, S., Yuan, H., and Tao, S.: Spatial and temporal variations of AOD over land at the global scale, International Journal of Remote Sensing, 33, 2097–2111, https://doi.org/10.1080/01431161.2011.605088, 2012.

Li, L., Che, H., Derimian, Y., Dubovik, O., Luan, Q., Li, Q., et al.: Climatology of Fine and Coarse Mode Aerosol Optical Thickness Over East and South Asia Derived From POLDER/PARASOL Satellite. Journal of Geophysical Research: Atmospheres, 125, e2020JD032665, https://doi.org/10.1029/2020JD032665, 2020.Li, T., Shen, H., Yuan, Q., and Zhang, L.: A locally weighted neural network constrained by global training for remote sensing estimation of $PM_{2.5}$, IEEE Transactions on Geoscience and Remote Sensing, 1–13, https://doi.org/10.1109/TGRS.2021.3074569, 2021.

O'Neill, N. T.: Comment on "Classification of aerosol properties derived from AERONET direct sun data" by Gobbi et al. (2007), Atmospheric Chemistry and Physics, 10, 10,017–10,019, https://doi.org/10.5194/acp-10-10017-2010, 2010.

O'Neill, N. T., Dubovik, O., and Eck, T. F.: Modified Ångström exponent for the characterization of submicrometer aerosols, Applied Optics, 40, 2368–2375, https://doi.org/10.1364/ao.40.002368, 2001a.

O'Neill, N. T., Eck, T. F., Smirnov, A., Holben, B. N., and Thulasiraman, S.: Spectral discrimination of coarse and fine mode optical depth, Journal of Geophysical Research: Atmospheres, 108, https://doi.org/10.1029/2002jd002975, 2003.

O'Neill, N., Eck, T., Smirnov, A., and Holben, B. Spectral deconvolution algorithm (SDA) technical memo, 2008.

O'Neill, N. T.: Comment on "Classification of aerosol properties derived from AERONET direct sun data" by Gobbi et al. (2007), Atmospheric Chemistry and Physics, 10, 10017-10019, 10.5194/acp-10-10017-2010, 2010. Ramachandran, S.: Aerosol optical depth and fine mode fraction variations deduced from Moderate Resolution Imaging Spectroradiometer (MODIS) over four urban areas in India, Journal of Geophysical Research: Atmospheres, 112, https://doi.org/10.1029/2007jd008500, 2007.

Sayer, A. M., Munchak, L. A., Hsu, N. C., Levy, R. C., Bettenhausen, C., and Jeong, M. J.: MODIS Collection 6 aerosol products: Comparison between Aqua's e-Deep Blue, Dark Target, and "merged" data sets, and usage recommendations, Journal of Geophysical Research: Atmospheres, 119, 13,965–913,989, https://doi.org/10.1002/2014JD022453, 2014.

Shen, H., Li, T., Yuan, Q., and Zhang, L.: Estimating regional ground-level $PM_{2.5}$ directly from satellite top-of-atmosphere reflectance using deep belief networks, Journal of Geophysical Research: Atmospheres, 123, 13,875–813,886, https://doi.org/10.1029/2018JD028759, 2018.

Wei, Y., Li, Z., Zhang, Y., Chen, C., Dubovik, O., Zhang, Y., Xu, H., Li, K., Chen, J., Wang, H., Ge, B., and Fan, C.: Validation of POLDER GRASP aerosol optical retrieval over China using SONET observations, Journal of Quantitative Spectroscopy and Radiative Transfer, 246, https://doi.org/10.1016/j.jqsrt.2020.106931, 2020.

Wei, Y., Li, Z., Zhang, Y., Chen, C., Xie, Y., Lv, Y., and Dubovik, O.: Derivation of $PM_{10}$ mass concentration from advanced satellite retrieval products based on a semi-empirical physical approach, Remote Sensing of Environment, 256, 112319, https://doi.org/10.1016/j.rse.2021.112319, 2021.

Yan, X., Li, Z., Shi, W., Luo, N., Wu, T., and Zhao, W.: An improved algorithm for retrieving the fine-mode fraction of aerosol optical thickness. Part 1: Algorithm development, Remote Sensing of Environment, 192, 87–97, https://doi.org/10.1016/j.rse.2017.02.005, 2017.

Yan, X., Zang, Z., Liang, C., Luo, N., Ren, R., Cribb, M., and Li, Z.: New global aerosol fine-mode fraction data over land derived from MODIS satellite retrievals, Environmental Pollution, 276, https://doi.org/10.1016/j.envpol.2021.116707, 2021b.

Zhang, Y. and Li, Z.: Remote sensing of atmospheric fine particulate matter ($PM_{2.5}$) mass concentration near the ground from satellite observation, Remote Sensing of Environment, 160, 252–262, https://doi.org/10.1016/j.rse.2015.02.005, 2015.

Zhang, Y., Li, Z., Qie, L., Zhang, Y., Liu, Z., Chen, X., Hou, W., Li, K., Li, D., and Xu, H.: Retrieval of aerosol fine-ode fraction from intensity and polarization measurements by PARASOL over East Asia, Remote Sensing, 8, https://doi.org/10.3390/rs8050417, 2016.

Zhang, Y., Li, Z., Liu, Z., Wang, Y., Qie, L., Xie, Y., Hou, W., and Leng, L.: Retrieval of aerosol fine-mode fraction over China from satellite multiangle polarized

observations: validation and comparison, Atmos. Meas. Tech., 14, 1655-1672, https://doi.org/10.5194/amt-14-1655-2021, 2021.

---

## Author Comment (AC3)

**Responses to RC3:**

In this manuscript, the authors developed and presented the new dataset of fine-mode fraction over global land during 2001-2020. The method they proposed is a hybrid physical and deep learning method, which is the physical model output calibration with DL. Generally, this FMF dataset can be useful for studies of anthropogenic aerosol and also the PM2.5 estimation. The paper is written in a consistent workflow and the inter-comparisons in terms of different methods and official products are very comprehensive. However, I think some concerns and issues need to be addressed.

**Response**: Thank you very much for your constructive suggestions to our manuscript.

Major comments:
1. We know that in deep learning modeling, the test dataset should be independent of the training dataset to avoid the data leakage, therefore my first concern is that the validation for FMF is independent. In Figure 3a, the authors use the AERONET data for training and testing. A more rigorous validation should be added. My suggestion is to conduct independent validation by FMF from other sources of FMF observations, or the Phy-DL FMF is only reliable over the pixels with AERONET sites.

**Response**: Thank you for this valuable recommendation. We have conducted a validation based on measurements from four independent National Oceanic and Atmospheric Administration Surface Radiation Budget (SURFRAD) network sites not used for training in the deep-learning model. The SURFRAD network provides long-term, multi-band AOD observations at a temporal resolution of three minutes. As shown in **Figure R1**, the four sites (black triangles) are located across the US, covering different land types from forested land to barren land (**Tables R1 and R2**). The land types were based on MODIS MCD12C1 data from the International Geosphere-Biosphere Programme scheme.

[Figure]

**Figure R1**. (a) Locations of AERONET sites (green dots) and four independent SURFRAD sites (black triangles, **Table R1**) for the independent validation of the Phy-DL FMF algorithm. The base map shows the land types from MODIS MCD12C1 data

(the International Geosphere-Biosphere Programme scheme) in 2010. **Table R2** provides details about the land-type legend.

**Table R1**. SURFRAD sites used for out-of-site validation and their locations and land types.

| Sites | Longitude (°W) | Latitude (°N) | Land type |
|---|---|---|---|
| Desert Rock (DRA) | 116.02 | 36.62 | Barren or sparse |
| Fort Peck (FPK) | 105.10 | 48.31 | Grasslands |
| Goodwin Creek (GWN) | 89.87 | 34.25 | Woody savannas |
| Penn State (PSU) | 77.93 | 40.72 | Mixed forests |

**Table R2**. Land types and corresponding values from MODIS MCD12C1 data (the International Geosphere-Biosphere Programme scheme).

| Value | Land type | Value | Land type |
|---|---|---|---|
| 1 | Evergreen needleleaf | 9 | Savannas |
| 2 | Evergreen broadleaf | 10 | Grasslands |
| 3 | Deciduous needleleaf | 11 | Permanent wetlands |
| 4 | Deciduous broadleaf | 12 | Croplands |
| 5 | Mixed forests | 13 | Urban and built up |
| 6 | Closed shrubland | 14 | Crop natural vegetation mosaic |
| 7 | Open shrublands | 15 | Snow and ice |
| 8 | Woody savannas | 16 | Barren or sparse |

We used the same AERONET method (SDA) to calculate the FMF at the SURFRAD sites. SURFRAD data were not included in the model training, so SURFRAD FMFs can be regarded as the out-of-site validation for the Phy-DL FMF algorithm. **Figure R2** shows how SURFRAD and Phy-DL FMFs compare. The correlation coefficient (R) was 0.51, and the root-mean-square error (RMSE) was 0.144, somewhat poorer performance than AERONET validation results (i.e., R=0.68 and RMSE=0.136).

[Figure]

**Figure R2**. Phy-DL FMF at 500 nm as a function of SURFRAD FMF. The black and red solid lines are the 1:1 line and the best-fit line obtained from linear regression, respectively. The black dashed and dotted lines represent the expected error (EE) envelopes of ±20% and ±40%, respectively.

Furthermore, the Phy-DL FMF performance was validated at each SURFRAD site. **Figure R3** shows the bias of Phy-DL FMF (Phy-DL FMF minus SURFRAD FMF), retrievals falling within the ±20% EE envelope, and RMSEs at each SURFRAD site. In general, most of the sites have a mean bias and an RMSE lower than 0.1 and 0.15, respectively, with over 70% of the retrievals falling within the ±20% EE envelope. The out-of-site validation reveals that the Phy-DL FMF algorithm is reliable even in regions without AERONET sites for model training.

[Figure]

**Figure R3**. Boxplots of bias (Phy-DL FMF minus SURFRAD FMF), percentage of FMF estimates falling within the EE envelope of ±20% (dash-dotted line), and RMSEs at the four independent SURFRAD sites. The upper, middle, and lower lines in each box present the 75th, median, and 25th percentiles, respectively. The red point in each box represents the mean value of the FMF bias.

2. My second concern is the use of meteorological data, which are very different inputs compared to previous studies. I can see the meteorological data are widely used in fine particulate (PM2.5) retrievals because meteorology has statistical correlation and physical interaction with the PM2.5. While in this study, the author simply explained their reason as "the impact of meteorological factors", and what is this "impact" to make them use the meteorological data is not mentioned. For example, the temperature is introduced as input, I don't see the influence of temperature on FMF or how it can improve the retrieval accuracy.

**Response**: Thank you for this question. First, the impact of meteorological factors revealed in the significant statistical correlation between meteorological factors and FMF. We implemented the Generalized Additive Model (GAM) to fit the meteorological variables to the FMF and investigated their nonlinear relationships. **Figure R4** reveals that the meteorological variables considered, i.e., PBLH, temperature, surface pressure, RH, and wind speed, all had significant non-linear relationships with the FMF (at the 99% significance level). Both PBLH and surface pressure had similar influences on the FMF, i.e., a positive (negative) response when PBLH and surface pressure values were low (high). This is because high PBLH and surface pressure values can increase the diffusion of fine particles, decreasing the magnitude of the FMF (Tai et al., 2010). Meanwhile, the negative response of the FMF to wind speed also reflects the influence of fine-particle diffusion, as well as the contribution of dust particles strengthened by wind speed (Luo et al., 2016). Increasing temperatures corresponded to decreasing FMFs, partly due to unfavorable diffusion conditions (Tai et al., 2010). On the other hand, more fine particles are released by heating during colder seasons than during warmer seasons (Ramachandran, 2007). RH had a strong positive influence on $PM_{2.5}$ concentrations when RH was between 25% and 75%. This reflects the secondary particle formation boosted by the increasing RH, contributing to the fine particles (Tai et al., 2010).

[Figure]

**Figure R4**. GAM fitting plots for the meteorological variables and the FMF. Shaded areas in the GAM plots indicate 95% confidence intervals, and the y-axes show the

covariate and effective degrees of freedom of the smoothing. Asterisks (**) after each p-value indicate the 99% confidence interval of fitting.

In addition, the physical approach (SDA) does not include these potential meteorological influences. O'Neill et al. (2008) have reported that when the temperature is low, the error of the fine-mode AOD calculated by the SDA is clearly large (**Figure R5**). Although the developers of the SDA know about this issue, the relationship between meteorological factors and the FMF is complex, difficult to describe in equations. Therefore, we incorporate meteorological variables into deep learning to model these complex relationships with the FMF.

[Figure]

**Figure R5**. Variation in $\Delta\tau_f$ with detector temperature (PEARL CIMEL, May 1 to August 31, 2007). Copied from the Spectral Deconvolution Algorithm (SDA) technical memo.

Changes in the manuscript: We have revised the manuscript as follows:
(1) Added a new figure (i.e., **Figure S2)** in the supplementary file.

[Figure]

**Figure S2**. GAM fitting plots for the meteorological variables and the FMF. Shaded areas in the GAM plots indicate 95% confidence intervals, and the y-axes shows the covariate and effective degrees of freedom of the smoothing. The asterisks (**) after each p-value indicate the 99% confidence interval of fitting.

(2) In section 2.3 entitled "Meteorological data", we have added the following text:

Previous studies have reported that meteorological variables are significantly correlated to fine-mode and coarse-mode aerosols. Tai et al. (2010), Liang et al. (2016), and Shen et al. (2018) all revealed that meteorological variables like temperature, RH, and wind speed explain much of the variations in $PM_{2.5}$ concentrations (> 50%). Xiang et al. (2019) and Gui et al. (2019) found a negative association between BLH and $PM_{2.5}$, and Kang et al. (2014) found that fine-mode aerosols and air pressure were correlated significantly. In this study, to investigate the correlation between meteorological variables and the FMF, we implemented the Generalized Additive Model (GAM). Figure S2 reveals that the meteorological variables considered in this study, i.e., PBLH, temperature, surface pressure, RH, and wind speed, all had significant non-linear relationships with the FMF (at the 99% significance level). Both PBLH and surface pressure had similar influences on the FMF, i.e., a positive (negative) response when PBLH and surface pressure values were low (high). This is because high PBLH and surface pressure values can increase the diffusion of fine particles, decreasing the magnitude of the FMF (Tai et al., 2010). Meanwhile, the negative response of the FMF to wind speed also reflects the influence of fine-particle diffusion, as well as the contribution of dust particles strengthened by wind speed (Luo et al., 2016). Increasing temperatures corresponded to decreasing FMFs, partly due to unfavorable diffusion conditions (Tai et al., 2010). On the other hand, more fine particles are released by heating during colder seasons than during warmer seasons (Ramachandran, 2007). RH had a strong positive influence on $PM_{2.5}$ concentrations when RH was between 25% and 75%. This reflects the secondary particle formation boosted by the increasing RH that contributed to the fine particles (Tai et al., 2010). Therefore, in this study, we used surface temperature, air pressure, PBLH, RH, and wind speed as inputs to the deeplearning model.

3. For the physical model, although LUT-SDA has been used in other studies before, it is better for authors to emphasize its disadvantage with more details, rather than just listing its applications.

**Response**: Thank you for this valuable suggestion. Because LUT-SDA is derived based on SDA, which does not include these potential meteorological influences. While O'Neill et al. (2008) have reported that when the temperature is low, the error of the fine-mode AOD calculated by the SDA is clearly large (**Figure R5**). Although the developers of the SDA know about this issue, the relationship between meteorological factors and the FMF is complex, difficult to describe in equations.

In addition, the core of the SDA method relies on AE as input (Yan et al., 2017). The Ångström Exponent (AE) from the MODIS DT aerosol product is still highly uncertain, thus the low accuracy of AE can significantly influence the performance of the Phy-DL FMF algorithm.

**Changes in the manuscript:** We have revised the manuscript as follows:

In section 1 entitled "Introduction", we have added the following text:

Because LUT-SDA is derived based on SDA, which does not include these potential meteorological influences. While O'Neill et al. (2008) have reported that when the temperature is low, the error of the fine-mode AOD calculated by the SDA is clearly large. In addition, the core of the SDA method relies on AE as input (Yan et al., 2017). The Ångström Exponent (AE) from the MODIS DT aerosol product is still highly uncertain, thus the low accuracy of AE can significantly influence the performance of the Phy-DL FMF algorithm.

4. Last I think the uncertainty of this Phy-DL FMF should have a more in-depth discussion. The major content in Results is the inter-comparisons of different results in terms of different methods and FMF products, which are good, but there should be more discussion on the sources of uncertainty of this Phy-DL FMF. For example, the Figure 6 compared performance over different land types and barren land has the worst performance for all three FMFs, so the authors could discuss how the physical characteristics of barren land affected the retrieval performance.

**Response**: Thank you for this valuable suggestion. We discussed the sources of uncertainty of this Phy-DL FMF further.

As shown in **Figure R6,** more than 75% of the sites located on barren land have low percentages of Phy-DL FMFs (< 60%) falling within the ±20% EE envelope. About 4% of the sites have high percentages of Phy-DL FMFs (> 90%) falling within the ±20% EE envelope.

[Figure]

**Figure R6**. Bar plots of the percentage of sites with > 90% of retrievals falling within the ±20% EE envelope (blue bars) and the percentage of sites with < 60% of retrievals falling within the ±20% EE envelope (red bars) for five land types.

The barren land type is a bright surface compared to other land types where the other sites are located (**Table R1**). AODs over the bright surface used for the Phy-DL FMF retrieval were significantly overestimated, with the worst performance compared to other vegetated land-cover types (Levy et al., 2010; Petrenko and Ichoku, 2013). This suggests that the performance of the Phy-DL FMF algorithm is poor when applied to regions with barren land. **Figure R7** shows the bias of the Phy-DL FMF and the percentage of retrievals falling within the EE envelope of ±20% as a function of NDVI. As NDVI increased from < 0.1 to > 0.8, the percentage of FMF retrievals falling within the ±20% EE envelope also rose from < 70% to > 85%, and the range of bias decreased significantly. Because a higher NDVI value indicates a darker surface, **Figure R7** reveals that the Phy-DL FMF algorithm performs better over dark surfaces than over bright surfaces, resulting in a lower accuracy over the barren land type than vegetated land types.

[Figure]

**Figure R7**. Box plots of the FMF bias (estimated FMF minus AERONET FMF) as a function of NDVI. The black horizontal dashed line indicates the zero bias. The gray dot in each box represents the mean value of the FMF bias. The upper, middle, and lower horizontal lines in each box show the 75th, median, and 25th percentiles, respectively. The green dots connected by the dashed curve are percentages of FMF retrievals falling within the EE envelope of ±20%.

The Ångström Exponent (AE) from the MODIS DT aerosol product is also a source of uncertainty in Phy-DL FMF. The core of the SDA method relies on AE as input (Yan et al., 2017), thus the low accuracy of AE can significantly influence the performance of the Phy-DL FMF algorithm. As shown in **Figure R8**, AEs from the MODIS MOD08 product used as input to the Phy-DL FMF algorithm performed the worst over barren land, with the highest RMSE (> 1) and the lowest percentage of retrievals falling within the EE envelope of ±0.45 (< 45%). This would result in a lower performance of the Phy-DL FMF algorithm when applied to regions with barren land.

[Figure]

**Figure R8**. RMSEs (bars) and percentages of MOD08 AE falling within the EE envelope of ±0.45 (dash-dotted line) against AERONET observations for five land types. The EE envelope (±0.45) was adopted from Levy et al. (2013).

Changes in the manuscript: We have revised the manuscript as follows:
(1) Added new figures (i.e., **Figure S3** and **Figure S4)** to the supplementary file.

[Figure]

**Figure S3**. (a) Bar plots of the percentage of sites with > 90% of retrievals falling within the ±20% EE envelope (blue bars) and the percentage of sites with < 60% of retrievals falling within the ±20% EE envelope (red bars) for five land types. (b) Box plots of the FMF bias (estimated FMF minus AERONET FMF) as a function of NDVI. The black horizontal dashed line indicates the zero bias. The gray dot in each box represents the mean value of the FMF bias. The upper, middle, and lower horizontal lines in each box show the 75th, median, and 25th percentiles, respectively. The green dots connected by the dashed curve are percentages of FMF retrievals falling within the EE envelope of ±20%.

[Figure]

**Figure S4**. RMSEs (bars) and percentages of MOD08 AE falling within the EE envelope of ±0.45 (dash-dotted line) against AERONET observation for five land types. The EE envelope (±0.45) was adopted from Levy et al. (2013).

(2) In section 3.1 entitled "Phy-DL FMF validation", we have added the following text:

Figure S3a shows that more than 75% of the sites located on barren land have low percentages of Phy-DL FMFs (< 60%) falling within the EE envelope of ±20%. About 4% of the sites have high percentages of Phy-DL FMFs (> 90%) falling within the ±20% EE envelope. This suggests that the accuracy of Phy-DL FMF over barren land is much lower than over other land types. The barren land type is a bright surface compared to other land types where the other sites are located (Table S3). AODs over the bright surface used for the Phy-DL FMF retrieval were significantly overestimated, with the worst performance compared to other vegetated land-cover types (Levy et al., 2010; Petrenko and Ichoku, 2013). This suggests that the performance of the Phy-DL FMF algorithm is poor when applied to regions with barren land. Figure S3b shows the bias of the Phy-DL FMF and the percentage of retrievals falling within the EE envelope of ±20% as a function of NDVI. As NDVI increased from < 0.1 to > 0.8, the percentage of FMF retrievals falling within the ±20% EE envelope also rose from < 70% to > 85%, and the range of bias decreased significantly. The Ångström Exponent (AE) from the MODIS DT aerosol product is still highly uncertain. The core of the SDA method relies on AE as input (Yan et al., 2017). The low accuracy of AE can significantly influence the performance of the Phy-DL FMF algorithm. As shown in Figure S4, AEs from the MODIS MOD08 product used as input to the Phy-DL FMF algorithm performed the worst over barren land, with the highest RMSE (> 1) and the lowest percentage of retrievals falling within the EE envelope of ±0.45 (< 45%). This would result in a lower performance of the Phy-DL FMF algorithm when applied to regions with barren land.

Minor comments:

1. in Line 44. The "performed previously,; currently" should be "performed previously; currently"
**Response**: Thank you for this correction. We have corrected it in Line 44 as "performed previously; currently".

2. Figure 1. AATSR and VIIRS also provide FMF products but they were not discussed in this paper. The reason for using MODIS rather than AATSR and VIIRS should also be mentioned.
**Response**: Thank you for this suggestion. We added the reasons in Line 50-51 as:
In addition, Advanced Along Track Scanning Radiometer (AATSR) ended the mission in 2012 (Kolmonen et al., 2016). While VIIRS started the mission in 2012, which could provide less than 10-year global FMF product so far (Sawyer et al., 2020).

3. in Line 56. The Yonsei Aerosol Retrieval algorithm (Choi et al., 2016) is not for MODIS land-based FMF retrievals, it is for GOCI.
**Response**: Thank you for this correction. We deleted this irrelevant description in Line 56.

4. in Line 103. There is no direct relative humidity data ERA5. Usually it is calculated

from dew point temperature.
Response: Thank you for this correction. The relative humidity was indeed calculated from dew point temperature and air temperature, and we corrected the "relative humidity" in Line 103 as "2-m dew point temperature" and added "the relative humidity (RH) was then calculated by 2-m dew point temperature and air temperature (Tetens, 1930)."

5. in Line 104. The "ERA5" mentioned for the first time without given the full name.
Response: Thank you for this correction. We actually have explained its full name in Line 104-105 as "ERA5 is the fifth-generation product produced by the European Centre for Medium Range Weather Forecasts". And here we corrected the Line 104-105 as "obtained from the fifth-generation product produced by the European Centre for Medium Range Weather Forecasts (ERA5) (Figure S1b-f), with hourly data available since 1950 and at a 0.25° spatial resolution."

6. in Line 210. The "yr" mentioned for the first time without given the full name.
Response: Thank you for this correction. We added its full name in Line 210 as "year (yr)".

7. in 3.2. Both past and present tense showed in this part when describing the same thing. For example, "in India, FMFs are noticeably higher in autumn and winter, especially in Northern India (i.e., the Indo-Gangetic Plain), where the FMF was greater than 0.87", and they should be either past or present tense.
Response: Thank you for this correction. We thoroughly checked the tense in **3.2 Global land FMF spatial distribution and trends from 2001 to 2020** and corrected the description.

8. in Figure 6a. Why do you choose to compare the result in bins of FMF?
Response: Thank you for this question. This comparison followed the Levy et al. (2007) in **Figure R9**, which also compared different FMF products using the bins of FMF.

[Figure]

**Figure R9**. MODIS aerosol size retrievals compared with AERONET-derived products.

The solid shapes and error bars represent the means and standard deviations of the MODIS retrievals in 20 bins of AERONET-derived product. Retrievals from V5.1 (orange) and V5.2 (green) are shown. The regressions (solid lines) are for the clouds of all points (not shown). The η over land is retrieved at 0.55 μm and compared with AERONET η retrieved by the O'Neill et al. (2003) method. Note that η is defined differently for MODIS and AERONET and that we only show results for τ > 0.20. Copied from Levy et al. (2007).

9. in Line 318 and 320. "eastern China" or "Eastern China"?
**Response**: Thank you for this correction. We corrected the "Eastern China" in Line 318 as "eastern China".

10. I noticed the plurals appeared randomly, for example, it is "AERONET FMF" or "AERONET FMFs". Make sure the plurals are consistent in the paper.
**Response**: Thank you for this correction. We thoroughly checked plurals in the manuscript and corrected the description in the manuscript.

**References:**
Gui, K., Che, H., Wang, Y., Wang, H., Zhang, L., Zhao, H., Zheng, Y., Sun, T., and Zhang, X.: Satellite-derived $PM_{2.5}$ concentration trends over Eastern China from 1998 to 2016: relationships to emissions and meteorological parameters, Environmental Pollution, 247, 1125–1133, https://doi.org/10.1016/j.envpol.2019.01.056, 2019.

Kang, P., Feng, N., Wang, Z., Guo, Y., Wang, Z., Chen, Y., Zhan, J., Zhan, F. B., and Hong, S.: Statistical properties of aerosols and meteorological factors in Southwest China, Journal of Geophysical Research: Atmospheres, 119, 9914–9930, https://doi.org/10.1002/2014JD022083, 2014.

Kolmonen, P., Sogacheva, L., Virtanen, T. H., de Leeuw, G., and Kulmala, M.: The ADV/ASV AATSR aerosol retrieval algorithm: current status and presentation of a full-mission AOD dataset, International Journal of Digital Earth, 9, 545-561, 10.1080/17538947.2015.1111450, 2016.

Levy, R. C., Remer, L. A., Mattoo, S., Vermote, E. F., and Kaufman, Y. J.: Second-generation operational algorithm: retrieval of aerosol properties over land from inversion of Moderate Resolution Imaging Spectroradiometer spectral reflectance, Journal of Geophysical Research: Atmospheres, 112, https://doi.org/10.1029/2006jd007811, 2007.

Levy, R. C., Mattoo, S., Munchak, L. A., Remer, L. A., Sayer, A. M., Patadia, F., and Hsu, N. C.: The Collection 6 MODIS aerosol products over land and ocean, Atmospheric Measeurement Techniques, 6, 2989–3034, https://doi.org/10.5194/amt-6-2989-2013, 2013.

Liang, X., Li, S., Zhang, S., Huang, H., and Chen, S. X.: $PM_{2.5}$ data reliability, consistency, and air quality assessment in five Chinese cities, Journal of Geophysical Research: Atmospheres, 121, 10,220–10,236, https://doi.org/10.1002/2016JD024877, 2016.

Luo, N., An, L., Nara, A., Yan, X., and Zhao, W.: GIS-based multielement source analysis of dustfall in Beijing: a study of 40 major and trace elements, Chemosphere, 152, 123–131, 2016.

O'Neill, N. T., Eck, T. F., Smirnov, A., Holben, B. N., and Thulasiraman, S.: Spectral discrimination of coarse and fine mode optical depth, Journal of Geophysical Research: Atmospheres, 108, https://doi.org/10.1029/2002jd002975, 2003.

O'Neill, N., Eck, T., Smirnov, A., and Holben, B. (2008). Spectral deconvolution algorithm (SDA) technical memo.

Petrenko, M. and Ichoku, C.: Coherent uncertainty analysis of aerosol measurements from multiple satellite sensors, Atmospheric Chemistry And Physics, 13, 6777–6805, https://doi.org/10.5194/acp-13-6777-2013, 2013.

Ramachandran, S.: Aerosol optical depth and fine mode fraction variations deduced from Moderate Resolution Imaging Spectroradiometer (MODIS) over four urban areas in India, Journal of Geophysical Research: Atmospheres, 112, https://doi.org/10.1029/2007jd008500, 2007.

Sawyer, V., Levy, R. C., Mattoo, S., Cureton, G., Shi, Y., and Remer, L. A.: Continuing the MODIS Dark Target Aerosol Time Series with VIIRS, 12, 308, 2020.

Shen, L., Jacob, D. J., Mickley, L. J., Wang, Y., and Zhang, Q.: Insignificant effect of climate change on winter haze pollution in Beijing, Atmospheric Chemistry and Physics, 18, 17,489–17,496, https://doi.org/10.5194/acp-18-17489-2018, 2018.

Tai, A. P. K., Mickley, L. J., and Jacob, D. J.: Correlations between fine particulate matter ($PM_{2.5}$) and meteorological variables in the United States: implications for the sensitivity of $PM_{2.5}$ to climate change, Atmospheric Environment, 44, 3976–3984, https://doi.org/10.1016/j.atmosenv.2010.06.060, 2010.

Tetens, V.O.: Uber einige meteorologische. Begriffe, Zeitschrift fur Geophysik. 6:297-309. 1930.

Xiang, Y., Zhang, T., Liu, J., Lv, L., Dong, Y., and Chen, Z.: Atmosphere boundary layer height and its effect on air pollutants in Beijing during winter heavy pollution, Atmospheric Research, 215, 305–316, https://doi.org/10.1016/j.atmosres.2018.09.014, 2019.

Yan, X., Li, Z., Shi, W., Luo, N., Wu, T., and Zhao, W.: An improved algorithm for retrieving the fine-mode fraction of aerosol optical thickness, part 1: Algorithm development, Remote Sensing of Environment, 192, 87-97, 10.1016/j.rse.2017.02.005, 2017.